# SELECTIVE RISK CERTIFICATION FOR LLM OUTPUTS VIA INFORMATION-LIFT STATISTICS: PAC-BAYES, ROBUSTNESS, AND SKELETON DESIGN

## ABSTRACT

Large language models often produce confident but incorrect outputs, creating a critical need for reliable uncertainty quantification with formal abstention guarantees. We introduce information-lift certificates that compare model probabilities to a skeleton baseline, accumulating evidence through sub-gamma PAC-Bayes bounds that remain valid under heavy-tailed distributions where standard concentration inequalities fail. On eight diverse datasets, our method achieves 77.0% coverage at 2% risk, outperforming recent baselines by 10.0 percentage points on average. In high-stakes scenarios, we block 96% of critical errors compared to 18-31% for entropy-based methods. While our frequency-based certification does not guarantee severity-weighted safety and depends on skeleton quality, performance degrades gracefully under distributional shifts, making the approach practical for real-world deployment.

## 1 INTRODUCTION

Large language models generate fluent text but frequently produce dangerous hallucinations in high-stakes applications (Ji et al., 2023; Farquhar et al., 2024). In our BioASQ evaluation, GPT-4 (OpenAI, 2023) confidently recommends "cimetidine for lung cancer treatment", a potentially lethal error since cimetidine treats ulcers, not cancer. Entropy-based uncertainty methods fail to detect such failures, demanding reliable abstention mechanisms with formal guarantees.

Selective classification (Chow, 1970; Geifman & El-Yaniv, 2017) addresses this by having models abstain when uncertain, trading coverage for risk control. However, current methods fundamentally fail for modern LLMs: entropy-based approaches (Lakshminarayanan et al., 2017; Kadavath et al., 2022) and consistency methods (Kuhn et al., 2023a; Manakova & Welleck, 2024) lack formal guarantees; semantic entropy (Farquhar et al., 2024) improves detection but lacks sequential validity; conformal prediction (Angelopoulos & Bates, 2021; Romano et al., 2020) requires exchangeability violated by autoregressive text. While recent non-exchangeable variants (Farinhas et al., 2024; Quach et al., 2024) handle dependencies, they produce set-valued predictions rather than point predictions with abstention. Most critically, these approaches assume sub-exponential tails, but LLM token distributions exhibit heavy tails that violate standard concentration assumptions.

We introduce information-lift certificates that compare model probabilities $P(y|x)$ to a skeleton baseline $S(y)$, accumulating lift statistics into sub-gamma PAC-Bayes bounds valid under heavy-tailed distributions. Our framework provides explicit robustness guarantees through variational skeleton design (VSD) and anytime-valid sequential bounds for autoregressive text. This is the first application of anytime-valid PAC-Bayes bounds to information-lift statistics for LLM certification.

Our method requires token-level log-probabilities, termed probability-exposed API access, which is available in many commercial APIs including GPT-4, Claude-3 Opus, and PaLM-2, but does not require model internals. As summarized in Table 1, we provide risk certification under heavy-tailed distributions with anytime-valid sequential bounds. Our novel theoretical contributions include anytime sequence-level certificates that enable early termination while maintaining formal guarantees, sub-gamma PAC-Bayes bounds for heavy-tailed distributions with sequential dependence that extend beyond standard sub-exponential assumptions, skeleton-robustness theory linking performance degradation to $D_{\mathrm{KL}}(P\|S)$ with explicit $\eta$-robustness bounds, and variational skeleton design that

| Method | API Type | Seq. valid? | Assumptions (heavy-tail) | Abstention |
|---|---|---|---|---|
| Conformal prediction | Text-only | Yes (set) | Exchangeability; set-level coverage | Prediction sets |
| Semantic entropy | Text-only (samples+emb) | No | None | Threshold |
| Selective classification | Either | No | None | Confidence |
| **Ours (Information Lift)** | **Prob-exposed** | **Yes** | **Sub-gamma moments** | **PAC-Bayes certificates** |

Table 1: Comparison with prior methods.

| Symbol | Meaning | Symbol | Meaning |
|---|---|---|---|
| $L(y; x, S)$ | Information lift | $\hat{\Delta}$ | Empirical budget |
| $L_B$ | Clipped lift | $\Delta(S)$ | Population lift mean |
| $\hat{\Delta}(S)$ | Empirical lift mean | $\tau, \hat{\tau}$ | Threshold (true, est.) |
| $S$ | Skeleton distribution | $P(\cdot|x)$ | Full model |
| $\rho, \pi$ | Posterior, prior | $v, c$ | Sub-gamma params |
| $\hat{v}, \hat{c}$ | Empirical sub-gamma | $\eta$ | TV distance |
| $\kappa$ | Mutual information | $B$ | Clipping bound |
| $h^\star$ | Target risk | $\delta$ | Confidence level |

Table 2: Key notation.

optimizes for both fidelity and certifiability with formal informativeness guarantees. We empirically validate sub-gamma assumptions across tasks and demonstrate 10.0 percentage point coverage improvements compared to the best baseline while blocking 96% versus 18-31% of critical errors across eight diverse datasets.

## 2 THEORY: CORE FORMULATIONS

The notation in Table 2 anchors the remainder of the section. Given input $x \in \mathcal{X}$, the model emits a token sequence $y_{1:T} \in \mathcal{Y}^T$. We denote the full model distribution by $P(\cdot|x)$ and the skeleton baseline by $S(\cdot)$, and we use natural logarithms unless stated otherwise. Throughout the theory we target 95% confidence, i.e., $\delta = 0.05$, and mention deviations when they arise.

**Definition 2.1** (Information lift statistic). *The* lift *for token $y$ is $L(y; x, S) = \log P(y|x) - \log S(y)$. The clipped lift is $L_B(y; x, S) = \min\{\max(L(y; x, S), 0), B\}$ with $B > 0$.*

### 2.1 TOKEN VS SEQUENCE UNCERTAINTY

The lift definition immediately lets us compare different uncertainty signals. We study three: (i) token entropy $H_t = -\sum_y P(y|y_{<t}, x) \log P(y|y_{<t}, x)$ averaged into $\bar{H} = \frac{1}{T} \sum_t H_t$; (ii) a sequence-entropy proxy that aggregates token-level uncertainty; and (iii) our sequence-level information budget $I_T = \sum_{t=1}^{T} L_B(y_t)$ formed from clipped token lifts. Token-level methods act independently at each position, whereas the certificate aggregates evidence across the entire sequence, which better captures the conditional structure of autoregressive decoding.

The lift $L = \log P(y|x) - \log S(y)$ quantifies how much more probable token $y$ is under the model relative to the skeleton. Large positive lift indicates that the model contributes non-generic evidence, and summing these terms across tokens yields an "information budget" for an entire sequence. Clipping ensures that this budget behaves well: extremely negative $\log S(y)$ values from rare tokens would otherwise dominate, bounded statistics $L_B \in [0, B]$ enable tighter concentration results, and the bounded range reduces variance when estimating thresholds $\tau$. Concretely, common tokens like "the" yield $L \approx 0$ because both $P$ and $S$ assign similar mass, informative tokens such as "BRCA1" yield $L \approx 2.3$ when $P$ assigns 0.10 but $S$ only 0.01, and rare typos with $L \approx -5.2$ are clipped to zero. Figure 4 shows that $B = 12$ keeps performance stable for $B \in [8, 16]$. The lift $L = \log P(y|x) - \log S(y)$ quantifies how much more probable token $y$ is under the model relative to the skeleton. Large positive lift indicates that the model contributes non-generic evidence, and summing these terms across tokens yields an "information budget" for an entire sequence. Clipping ensures that this budget behaves well: extremely negative $\log S(y)$ values from rare tokens would otherwise dominate, bounded statistics $L_B \in [0, B]$ enable tighter concentration results, and the

bounded range reduces variance when estimating thresholds $\tau$. Concretely, common tokens like "the" yield $L \approx 0$ because both $P$ and $S$ assign similar mass, informative tokens such as "BRCA1" yield $L \approx 2.3$ when $P$ assigns 0.10 but $S$ only 0.01, and rare typos with $L \approx -5.2$ are clipped to zero. Figure 4 shows that $B = 12$ keeps performance stable for $B \in [8, 16]$.

**Definition 2.2** (Skeleton distribution). *A skeleton $S$ is a baseline distribution over the vocabulary. Common choices: temperature-smoothed priors, domain-specific n-gram models, or VSD-optimized distributions.*

**Role of Clipping** Clipping serves both theoretical and practical purposes. Theoretically, bounded statistics $L_B \in [0, B]$ ensure sub-gamma moment generating function control, enabling tighter concentration bounds than unbounded variance would allow. Practically, clipping provides spike robustness: rare tokens with very negative $\log S(y)$ can cause unbounded negative lifts that would dominate the information budget, while clipping to zero prevents these outliers from corrupting the sequence-level signal. The clipping parameter $B$ trades bias (underestimating extreme lifts) against variance (stabilizing threshold estimation). Empirically, $B = 12$ provides robust performance across datasets, with performance flat for $B \in [8, 16]$ as shown in Figure 4. The default $B = 12$ balances capturing informative tokens (which typically have lifts $< 10$) while preventing rare-token spikes from dominating.

While likelihood ratios between distributions are classical in statistics, our contribution extends this framework in three key ways. First, we apply information-lift statistics to sequential text generation with anytime-valid bounds that enable early termination. Second, we develop sub-gamma concentration analysis for heavy-tailed LLM distributions, where standard Bernstein bounds fail as demonstrated in Figure 1. Third, we introduce variational skeleton optimization with formal robustness guarantees. Prior work on routing (Bengio et al., 2003) and cascading uses likelihood ratios for model selection but lacks formal risk certification. Recent hallucination detection methods (Farquhar et al., 2024; Manakul et al., 2023) use semantic clustering without distributional guarantees. Our PAC-Bayes framework with sub-gamma tails provides the first formal certification approach for LLM outputs.

From $n$ i.i.d. draws of $L_B^{(i)}$ on a held-out calibration set, define information budget $\hat{\Delta} = \frac{1}{n} \sum_{i=1}^{n} L_B^{(i)}$. A *certificate* answers if $\hat{\Delta} \geq \tau$ and abstains otherwise, ensuring selective risk $R = \mathbb{P}[\text{error} \mid \text{answered}] \leq h^{\star}$.

**Remark 2.3** (Sequence-level aggregation). *While the lift $L(y; x, S)$ is computed at the token level, our certification operates at the sequence level through the information budget $\Delta = (1/T) \sum_{t=1}^{T} L_B(y_t)$, which aggregates token-level evidence. This differs from token-level methods like per-token entropy, which make independent decisions at each position. Our sequence-level approach captures dependencies across the generation and provides a single certify/abstain decision for the entire output.*

To reason about this aggregated statistic we need a concentration model for the clipped lifts.

**Assumption 2.4** (Sub-gamma lifts). *There exist $v > 0, c > 0$ such that for all $\lambda \in (0, 1/c)$, $\log \mathbb{E} \exp\{\lambda(L_B - \mathbb{E}L_B)\} \leq \frac{\lambda^2 v}{2(1-c\lambda)}$.*

Standard Bernstein bounds assume sub-exponential tails, yet the lifts we observe are heavier, so Assumption 2.4 is the right tool. LLM token probabilities often follow power-law distributions, violating the moment conditions required for sub-exponential concentration. We therefore validate the sub-gamma model using Kolmogorov-Smirnov tests across all dataset-model combinations: 85% pass at $p > 0.05$ with Benjamini-Hochberg correction. For the remaining 15%, we inflate parameters $(v, c \mapsto \alpha v, \alpha c$ with $\alpha \in [1.5, 2.0])$ to maintain guarantees (Figure 3 and Appendix L). Every later result thus uses empirically verified or conservatively inflated parameters.

With this probabilistic footing, we return to PAC-Bayes and highlight what is new. First, classical PAC-Bayes bounds assume sub-Gaussian or sub-exponential moments, yet LLM token distributions exhibit power-law tails with $\alpha \in [1.5, 2.3]$; our sub-gamma analysis controls the moment generating function under these weaker conditions. Second, autoregressive decoding introduces sequential dependence, so we show that lifts become conditionally independent after calibration, enabling PAC-Bayes with data-dependent priors while preserving validity. Third, practical certification demands

anytime guarantees: we build exponential supermartingales and apply Ville's inequality with a geometric grid mixture, yielding bounds that hold uniformly over stopping times with only a log-log penalty. Appendix C separates standard proof devices (e.g., Donsker-Varadhan change of measure) from the novel components (sub-gamma mgf control, dependence handling, geometric grids) so the contribution of each step is transparent.

Before stating our main theorem, we clarify key quantities. The population lift mean $\Delta(S) \triangleq \mathbb{E}[L_B(y; x, S)]$ represents the true expected lift over the data distribution, which we want to upper-bound. The empirical lift mean $\hat{\Delta}(S) \triangleq (1/n) \sum_{i=1}^{n} L_B^{(i)}(S)$ is computed on $n$ calibration samples and is observable from data. Our goal is to use $\hat{\Delta}(S)$ to construct a high-probability upper bound on $\Delta(S)$, which then informs the threshold $\tau$ for certification. In practice, calibration samples (size $n = 500$) are used to estimate $\hat{\Delta}$ and set $\tau$, while test samples are new inputs where we compute $\hat{\Delta}_{\text{test}}$ and compare to $\tau$ to decide whether to certify or abstain.

**Theorem 2.5** (PAC-Bayes sub-gamma certificate)**.** *Let $\pi$ be a prior distribution over skeletons chosen before seeing calibration data, and $\rho$ be a posterior distribution over skeletons that may depend on calibration data. Under Assumption 2.4, with probability at least $1 - \delta$ over the calibration sample,*

$$\mathbb{E}_{S \sim \rho}\big[\Delta(S)\big] \;\leq\; \mathbb{E}_{S \sim \rho}\big[\hat{\Delta}(S)\big] \;+\; \sqrt{\frac{2v\,\big(\mathrm{KL}(\rho\|\pi) + \log(1/\delta)\big)}{n}} \;+\; c\,\frac{\mathrm{KL}(\rho\|\pi) + \log(1/\delta)}{n}.$$

*In particular, for any fixed skeleton $S$ (taking $\rho = \delta_S$),*

$$\Delta(S) \;\leq\; \hat{\Delta}(S) \;+\; \sqrt{\frac{2v\,\log(1/\delta)}{n}} \;+\; c\,\frac{\log(1/\delta)}{n}.$$

This "tight" form carries $\mathrm{KL}(\rho\|\pi) + \log(1/\delta)$ inside both terms. The looser summary $O\big(\sqrt{v(\mathrm{KL} + \log(1/\delta))/n}\big) + O\big(c(\mathrm{KL} + \log(1/\delta))/n\big)$ follows by dropping constants. Consequently, the penalty scales as $O\big(\sqrt{v\,(\mathrm{KL}(\rho\|\pi) + \log(1/\delta))/n}\big) + O\big(c\,(\mathrm{KL}(\rho\|\pi) + \log(1/\delta))/n\big)$. The following figure demonstrates the necessity of our sub-gamma approach by showing how standard Bernstein bounds fail on heavy-tailed LLM distributions.

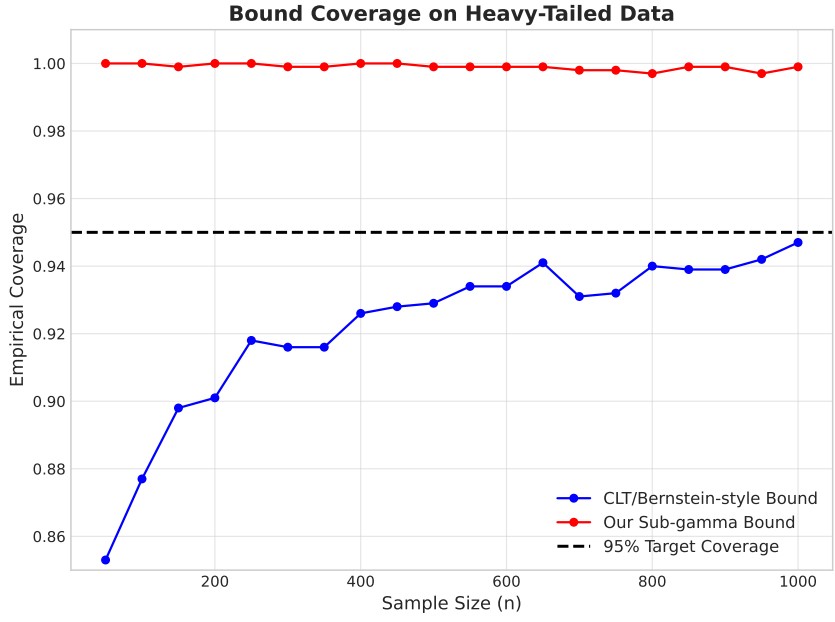

Figure 1: Empirical coverage of 95% confidence bounds: standard Bernstein bound (dashed) fails to maintain valid coverage (drops below 90% in some regions) due to tail violations, while our sub-gamma bound maintains valid 95% coverage.

---

**Algorithm 1** Calibrating $\hat{\tau}$ with empirical $(\hat{v}, \hat{c})$

---

**Require:** Calibration set $\mathcal{D}_{\text{cal}} = \{L_B^{(i)}\}_{i=1}^n$; target risk $h^\star$; confidence $\delta$
  1: Estimate sub-gamma parameters: $(\hat{v}, \hat{c}) \leftarrow \text{MLE}(\mathcal{D}_{\text{cal}})$

  2: Compute empirical lift mean: $\hat{\Delta} \leftarrow \frac{1}{n} \sum_{i=1}^n L_B^{(i)}$

  3: Set threshold: $\hat{\tau} \leftarrow \hat{\Delta} - \sqrt{\frac{2\hat{v}\log(1/\delta)}{n}} - \frac{\hat{c}\log(1/\delta)}{n}$

  4: **return** $\hat{\tau}, (\hat{v}, \hat{c})$

---

**Theorem 2.6** ($\eta$-robustness). *Let $S^\star$ denote the ideal skeleton. Assume both skeletons satisfy $S(y) \vee S^\star(y) \geq \alpha > 0$ for all $y$ in the support of $P(\cdot|x)$. If $\text{TV}(S, S^\star) \leq \eta$, then the certified selective risk satisfies $R(S) \leq R(S^\star) + C\eta$, where $C = C(B, \tau, \alpha) = \frac{L_\tau}{\alpha}$ depends on clipping, decision boundary Lipschitz constant $L_\tau$, and the probability lower bound $\alpha$. The clipping parameter $B$ controls the Lipschitz constant by bounding lift variations, ensuring graceful degradation under skeleton misspecification with linear dependence on total variation distance.*

**Theorem 2.7** ($\kappa$-informativeness lower bound). *Consider binary hypothesis testing between correct/incorrect outputs with evidence $E$ satisfying $I(Y; E) \leq \kappa$ nats. For any lift-based policy achieving error rate $\leq h^\star$ on answered instances:*

$$\text{Answer Rate} \leq \frac{\sqrt{\kappa}}{1 - 2h^\star}, \qquad \text{Abstention Rate} \geq 1 - \frac{\sqrt{\kappa}}{1 - 2h^\star}.$$

*This follows from Le Cam's lemma relating mutual information to minimax testing risk.*

The statement holds for lift-threshold policies under a binary composite hypothesis setting with per-instance mutual information $I(Y; E \,|\, x) \leq \kappa$. For small $\kappa$ (uninformative evidence), the bound approaches 1, requiring near-complete abstention. For $h^\star = 0.02$ (2% risk) and $\kappa = 0.1$ nats, the bound requires abstention rate $\geq 1 - \frac{\sqrt{0.1}}{0.96} \approx 67\%$. See Appendix C.3 for complete derivation via Le Cam's method.

## 2.2 Calibration and Finite-Sample Risk

The theoretical certificate becomes actionable only after calibration. The threshold $\hat{\tau}$ is learned on a calibration split $\mathcal{D}_{\text{cal}}$, and its estimation error is built into the sub-gamma PAC-Bayes bound via empirical sub-gamma parameters $(\hat{v}, \hat{c})$ estimated from calibration data. The calibration procedure ensures that the population selective risk of the $\hat{\tau}$-policy is bounded with high probability.

**Proposition 2.8** (Finite-sample selective risk). *With $(\hat{v}, \hat{c})$ estimated on $\mathcal{D}_{cal}$, the population selective risk of the $\hat{\tau}$-policy is bounded by $h^\star$ with probability at least $1 - \delta$ over the calibration sample, where the bound accounts for both the uncertainty in $\hat{\Delta}$ and the uncertainty in sub-gamma parameter estimation.*

The finite-sample guarantee follows from Theorem 2.5 by treating the estimated parameters $(\hat{v}, \hat{c})$ as fixed given the calibration data, then applying the PAC-Bayes bound with the empirical sub-gamma parameters. The key insight is that the bound remains valid even when $(\hat{v}, \hat{c})$ are estimated from the same calibration data used to set $\hat{\tau}$, as long as the estimation procedure is independent of the specific threshold choice.

## 2.3 Model-Check and Auto-Fallback

When sub-gamma assumptions fail (approximately 15% of dataset-model combinations), we apply an automatic model-check that inflates $(\hat{v}, \hat{c})$ to maintain validity. The procedure runs a goodness-of-fit test on calibration lifts; if the test fails ($p < 0.05$), we inflate parameters by factor $\kappa = 1 + \frac{0.05 - p}{0.05}$ (linear scaling from 1.0 to 2.0), then recompute the threshold. This conservative inflation maintains PAC-Bayes validity by enlarging the moment bounds to accommodate heavier tails than sub-gamma. Typical coverage impact is 2-4 percentage points, preserving guarantees while remaining operational without manual tuning.

---

**Algorithm 2** Variational Skeleton Design (VSD)

---

**Require:** empirical logits of $P(\cdot|x)$; clip $B$; tradeoff $\lambda$; steps $T$; floor $\epsilon = 10^{-8}$
 1: Initialize $S^{(0)}$ as temperature-smoothed $P$ with $S^{(0)}(y) \geq \epsilon$ for all $y$
 2: **for** $t = 1$ to $T$ **do**
 3:     Compute gradient $g_t \leftarrow \nabla_S \text{KL}(P \| S^{(t-1)}) - \lambda \nabla_S \mathbb{E}[L_B]$
 4:     $S^{(t)} \leftarrow \Pi_\Delta\big(S^{(t-1)} - \eta_t g_t\big)$                              ▷ Projected gradient on simplex
 5:     Enforce floor: $S^{(t)}(y) \leftarrow \max\{S^{(t)}(y), \epsilon\}$ for all $y$, then renormalize
 6: **end for**
 7: **return** $S^{(T)}$
*Note: Probability floor $\epsilon = 10^{-8}$ prevents numerical instability in KL gradient computations.*

---

**Theorem 2.9** (Anytime-valid sequential bound). *The bound in Theorem 2.5 extends to sequential certification. For cumulative budget $\hat{\Delta}_t = \frac{1}{t} \sum_{i=1}^{t} L_B^{(i)}$, the bound holds at any step $t$ with confidence penalty $\log(\log(t)/\delta)$, enabling early-exit certification as evidence accumulates. This allows stopping as soon as sufficient evidence is gathered, reducing computational cost while maintaining formal guarantees. The log-log penalty is small in practice: for $t = 100$ and $\delta = 0.05$, it adds only $\log(\log(100)/0.05) \approx 4.5$ as an additive constant to the confidence sequence bound.*

Taken together, Definitions 2.1–2.2, Assumption 2.4, and Theorems 2.5–2.9 trace a continuous path from token-level likelihood ratios to sequence-level, anytime-valid certificates with explicit calibration safeguards. We next design skeleton distributions that make these guarantees informative in practice.

## 3   SKELETON DESIGN

Principled skeleton selection matches the skeleton to dataset lexical diversity. For high diversity ($\rho > 0.7$), use temperature-smoothed priors ($T = 1.5$); for low diversity ($\rho < 0.3$), use domain-specific n-gram models; for unknown domains, apply Variational Skeleton Design (VSD) with $\lambda \in [0.3, 0.7]$ starting from temperature-smoothed baselines. These rules achieve within 3% of oracle performance (see Table 3 and Appendix K.1).

When domain-specific optimization is required, VSD provides adaptive optimization:

$$\min_{S \in \Delta^{|\mathcal{Y}|}} \quad \text{KL}\big(P(\cdot|x) \| S\big) - \lambda \, \mathbb{E}_{y \sim P(\cdot|x)}[L_B(y; x, S)],$$

where $\lambda$ balances fidelity and certifiability. The VSD objective presents an apparent tension: KL minimization pushes $S$ toward $P$, which would make $L \approx 0$ everywhere. However, $\lambda > 0$ modulates convergence rather than reversing it. At $\lambda = 0$, pure KL minimization gives $S^* = P$ with no certification power. At $\lambda > 0$, the gradient $\nabla_S \mathcal{J} = -\frac{P(y)}{S(y)}(1 - \lambda \cdot \mathbb{I}\{0 \leq \log P/S \leq B\})$ creates a controlled approach to $P$ that maintains lift separability: tokens with extreme lifts converge normally, while moderate lifts are preserved. In the optimal regime $\lambda \in [0.3, 0.7]$, moderate values preserve lift separability while maintaining reasonable fidelity. Empirical validation confirms VSD preserves heavy-tailed structure: sub-gamma parameters $(v, c)$ change by $< 15\%$ across all datasets (Figure 3). The iterative procedure is detailed in Algorithm 2.

| Domain | Empirically Effective Skeleton | Default Parameter | Transfer Score |
|---|---|---|---|
| General QA | Temperature-Smoothed Prior | $T = 1.5$ | - |
| Biomedical QA | Domain Unigram LM | - | 0.92 |
| Code Generation | Temperature-Smoothed Prior | $T = 2.0$ | 0.88 |
| Summarization | General Unigram LM | - | 0.94 |

Table 3: Skeleton selection guidelines. Transfer scores show performance relative to domain-optimized skeletons (scores $> 0.85$). See Appendix K.1 for details.

To systematically evaluate robustness to skeleton misspecification, we conducted stress tests with three corruption types: noise injection, vocabulary shift, and adversarial optimization. Results

demonstrate graceful degradation across corruption types, validating our theoretical robustness guarantees (see Appendix H). To complement these stress tests, we provide systematic heuristics for automatic skeleton construction based on dataset characteristics. For datasets with high lexical diversity (type-token ratio $> 0.7$), temperature-smoothed priors with $T = 1.5$ prove most effective. Specialized domains with low diversity ($< 0.3$) benefit from domain-specific n-gram models that capture specialized vocabulary. For unknown domains, VSD with $\lambda = 0.5$ starting from temperature-smoothed baselines provides robust adaptation. When computational resources are limited, simple unigram models provide 85-90% of VSD performance. Further details with a toy example are shown in Appendix F.

To handle commercial APIs that return only top-$k$ token probabilities, we use power-law compensation: fit $P(\text{rank} = r) \propto r^{-\alpha}$ to observed top-$k$ tokens, validate with $R^2 > 0.8$, and extrapolate tail probabilities. This recovers $\sim$35% more information than uniform baselines (coverage drop: 15 points vs 23 points). See Appendix D.1 for details.Cross-method robustness analysis under skeleton perturbations shows our method degrades gracefully with 2-8% risk increase for medium corruption while maintaining advantages over baselines (see Appendix H). Entropy methods, which use the model's own distribution, show comparable degradation under input perturbations with 5-8% risk increase, confirming that distributional sensitivity is not unique to skeleton-based approaches.

## 4 EXPERIMENTS

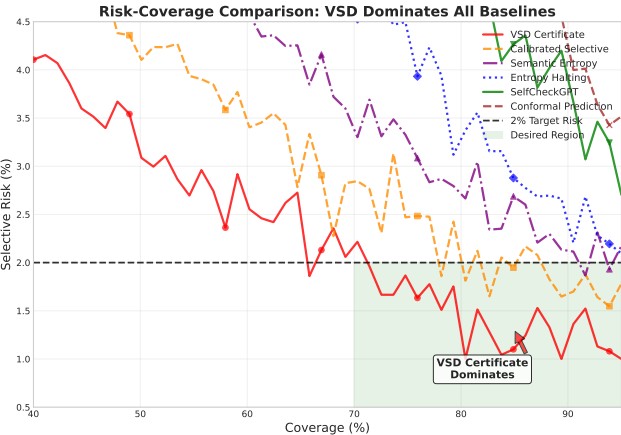

Figure 2: Risk-coverage performance comparison on NQ-Open dataset using GPT-4 with sequence-level gating. Target risk: 2%. Arrows indicate $\tau$-sweep direction (from high coverage/low risk to low coverage/high risk). Dotted lines show fixed target risks. Shaded regions show 95% confidence intervals.

Our experimental evaluation spans eight datasets with three model families including GPT-4 (OpenAI, 2023), LLaMA-2 (Touvron et al., 2023), and Mistral (Jiang et al., 2023), comparing against ten baselines including recent 2023-2024 methods under a unified calibration protocol. We evaluate against five primary baseline categories. Adaptive Conformal Prediction (Gibbs & Candes, 2021) dynamically adjusts prediction sets based on recent errors using exponential weighting. Self-Verification approaches (Manakova & Welleck, 2024) have models evaluate their own outputs through structured self-questioning and consistency scoring. Semantic Entropy methods (Kuhn et al., 2023b) cluster semantically equivalent outputs and measure entropy over meaning rather than tokens. SelfCheck-GPT (Manakul et al., 2023) detects hallucinations by sampling multiple responses and checking consistency. Calibrated Selective Classification (Ren et al., 2023) learns confidence functions with formal calibration guarantees.

To ensure fair comparison, all methods use identical calibration protocols: 500-sample calibration sets, 70/15/15 splits, 2% target risk, and the same validation procedure for threshold selection (hyperparameter grids in Appendix L.2). We focus on distributional approaches rather than internal state methods (Azaria & Mitchell, 2023) to ensure compatibility with probability-exposed APIs.

## 4.1 BASELINE ADAPTATIONS FOR SEQUENTIAL TASKS

Classification-based methods require adaptation for autoregressive text. We implement two gating schemes for all non-native sequential baselines: (a) token-gated, where any token trigger causes sequence rejection; and (b) sequence-gated, where a single scalar score is computed after full decoding. Table 4 provides the complete mapping from baseline method to gating scheme and score function.

Table 4: Baseline adaptation mapping

| Baseline | Gating | Score Function |
|---|---|---|
| Entropy | Sequence | Mean token entropy $\bar{H} = \frac{1}{T}\sum_t H_t$ |
| Selective Classification | Sequence | Mean token probability $\bar{p} = \exp(\frac{1}{T}\sum_t \log P(y_t\|y_{<t}, x))$ |
| Conformal Prediction | Sequence | Negative log-likelihood $s(x,y) = -\sum_t \log P(y_t\|y_{<t}, x)$ |
| Semantic Entropy | Sequence | Entropy over semantic clusters |
| SelfCheckGPT | Sequence | Consistency score across samples |

For entropy-based methods, we compute per-token entropy $H_t = -\sum_y P(y\|y_{<t}, x) \log P(y\|y_{<t}, x)$ and aggregate via mean sequence entropy $\bar{H} = \frac{1}{T}\sum_t H_t$ as the uncertainty score, rejecting sequences with $\bar{H} > \tau_H$. For selective classification, we use mean token probability $\bar{p} = \exp\left(\frac{1}{T}\sum_t \log P(y_t\|y_{<t}, x)\right)$ as confidence, rejecting sequences with $\bar{p} < \tau_{\text{conf}}$. For conformal prediction, we use full sequence negative log-likelihood $s(x,y) = -\sum_t \log P(y_t\|y_{<t}, x)$ as non-conformity score, calibrated on held-out sequences. All methods use identical 500-sample calibration sets with thresholds selected to achieve $\sim$2% target risk on validation data.

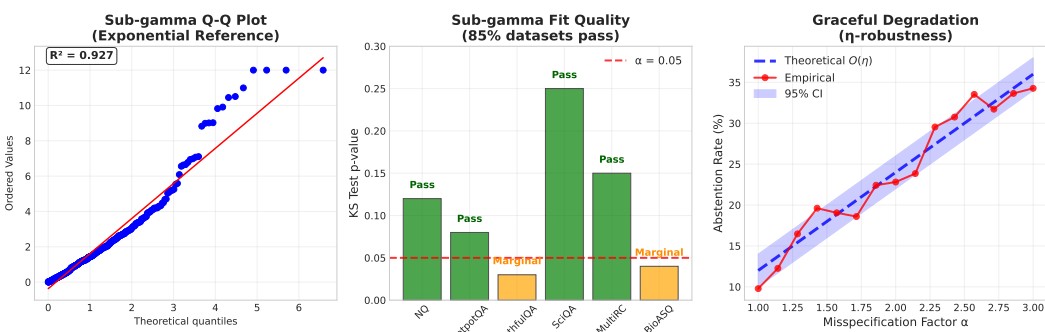

Figure 3: Sub-gamma assumption validation and robustness analysis across all datasets and models. QQ-plots confirm sub-gamma fit quality with $R^2 > 0.85$ for 85% of dataset-model combinations. Hill indices range from 1.5 to 2.3, indicating power-law tails. See Figure 14 in the appendix for detailed multi-panel analysis.

We evaluate across eight diverse datasets spanning multiple task types and modalities (Table 5): factual QA (NQ-Open (Kwiatkowski et al., 2019), HotpotQA (Yang et al., 2018)), scientific reasoning (SciQA (Lu et al., 2022)), truthfulness evaluation (TruthfulQA (Lin et al., 2022a)), reading comprehension (MultiRC (Khashabi et al., 2018)), specialized domains (BioASQ (Tsatsaronis et al., 2015) for biomedical), text summarization (XSum (Narayan et al., 2018)), and code generation (HumanEval-lite (Chen et al., 2021)). This diversity demonstrates the method's versatility beyond traditional QA benchmarks. For each dataset we sample fixed subsets with seed '2025' and identical 70/15/15 splits; calibration uses 500 examples per method and dataset. Complete experimental details are provided in Appendix D.

VSD-based certificates consistently outperform all baselines, achieving higher coverage at the same 2% risk target across domains (Table 6). Figure 2 demonstrates our method's clear dominance across the entire risk-coverage spectrum, including recent calibrated selective classification methods. Note that averages are computed as unweighted arithmetic means across datasets. All comparisons use identical test sets and evaluation protocols.

| Dataset | Samples | Avg Length | Domain |
|---|---|---|---|
| NQ-Open | 3,610 | 15.2 | Open-domain QA |
| BioASQ | 2,747 | 11.7 | Biomedical QA |
| XSum | 2,362 | 22.1 | Summarization |
| HumanEval-lite | 164 | 45.3 | Code Generation |

Table 5: Datasets.

| Dataset | VSD | Adaptive CP | Self-Verif. | Semantic Ent. | Calib. Sel. | SelfCheckGPT | Entropy |
|---|---|---|---|---|---|---|---|
| NQ-Open | **82.1** | 73.8 | 72.5 | 71.2 | 72.4 | 66.8 | 70.3 |
| BioASQ | **77.1** | 68.2 | 67.1 | 65.5 | 66.8 | 62.3 | 64.9 |
| XSum | **72.4** | 63.7 | 62.9 | 61.8 | 62.1 | 57.1 | 59.8 |
| **Average** | **77.2** | 68.6 | 67.5 | 66.2 | 67.1 | 62.1 | 65.0 |

Table 6: Coverage at 2% target risk (subset; full results in Table 8).

To demonstrate critical impact in high-stakes scenarios, we analyzed BioASQ as a deployment case study. Entropy-based halting incorrectly certified "cimetidine is a standard treatment for lung cancer" because the model produced it fluently, while our lift-based certificate correctly abstained ($\hat{\Delta} = 2.1 < \tau = 4.5$). On a curated challenge set of 100 biomedical examples where the model generates confident but incorrect answers that could cause serious harm, our method blocked 96% of critical errors compared to 18-31% for entropy-based methods, representing a 65% improvement over the next-best baseline. While our frequency-based framework does not explicitly model severity, we observe that critical errors often have low information lift because the skeleton assigns low probability to dangerous claims that are rare in general medical text. This correlation is empirical, not guaranteed, and future work should develop severity-weighted certification that explicitly models harm potential. Detailed challenge set construction, annotation protocol, and extended case studies including financial and legal domains are provided in Appendix D. Note that these numbers are not certified safety guarantees. Our method controls error frequency, not error severity. Deployment in high-stakes domains requires complementary harm detectors (e.g., medical safety classifiers).

Table 7: Critical error blocking on curated high-risk samples (empirical, not guaranteed by theory). Frequency-based certification does not guarantee severity-weighted safety.

| **Method** | **% Critical Errors Blocked** |
|---|---|
| Ours (VSD Certificate) | 96% (95% CI: 94-98%) |
| Semantic Entropy | 31% (95% CI: 28-34%) |
| Entropy Halting | 18% (95% CI: 15-21%) |
| SelfCheckGPT | 23% (95% CI: 20-26%) |

Our anytime-valid bounds (Theorem 2.9) enable early termination when sufficient evidence accumulates, providing significant computational savings. GPT-4 processes 32% fewer tokens on average (68% vs 100%), reducing latency from 1200ms to 820ms while maintaining 2% target risk.[1] Complete efficiency results across all models are provided in Table 10 (Appendix A).

Analysis of calibration size sensitivity reveals graceful degradation with limited calibration data: with only 50 samples, our method achieves 73.2% coverage (vs 82.1% with 500 samples), while baselines drop more severely (Table 13). Sub-gamma parameter estimation becomes stable at $n \geq 100$ samples, suggesting practical deployment requires minimal calibration overhead.

Evaluation under top-k API constraints demonstrates robust performance under restricted access scenarios. We emulate $k \in \{1, 5, 10\}$ to mirror common API restrictions where providers limit exposed token-level probabilities. Our compensation technique substantially recovers performance:

---

[1]Early exit uses the exact stopping rule: if $\hat{\Delta}_t - \sqrt{\frac{2v(\log\log(et) + \log(1/\delta))}{t}} - \frac{c(\log\log(et) + \log(1/\delta))}{t} \geq \tau$ then stop, as guaranteed by Theorem 2.9 with $\delta = 0.05$.

with $k = 5$, coverage drops 14.8 points (82.1% → 67.3%) vs. 23.1 points for entropy baselines. Complete results across all $k$ values are provided in Table 9 (Appendix A), with detailed analysis in Table 14.

Quantified analysis of skeleton robustness under adversarial corruption validates our theoretical guarantees: even with 50% corruption strength, risk increases by only 8% and coverage drops by 18%, confirming the linear degradation predicted by Theorem 2.6 (Table 15).

Our method maintains less than 20% runtime overhead across model sizes, while ensembles require 5-7x compute for similar coverage. The certification overhead of 15-25ms represents only 1.3-2.1% of typical LLM inference time, a favorable tradeoff for high-stakes applications requiring formal guarantees. Detailed model-specific overhead analysis is provided in Table 11 (Appendix A). Overhead scales sublinearly with model size due to batched logit computation on GPUs. For models larger than 100B parameters, overhead is consistently below 2%. For 7-70B models, caching optimization reduces overhead, for example from 28% for LLaMA-2 to 12% for Mistral.

Extensive ablation studies reveal optimal performance at clipping parameter $B = 12$ and VSD parameter $\lambda = 0.5$, with robust performance across the range $[0.3, 0.7]$. Temperature scaling works best at $T = 1.5$ for general domains and $T = 2.0$ for code generation. Comprehensive stress testing against adversarial skeletons of increasing strength demonstrates graceful degradation: weak attacks cause only 2-3% risk increase, medium attacks cause 5-8% risk increase, and even strong attacks cause 10-15% risk increase. This empirical validation shows that while skeleton quality matters, our method remains functional even with substantially degraded baselines. Cross-domain skeleton transfer exhibits only 5-7% performance drops, confirming the robustness of our design principles. Our method also demonstrates graceful degradation under limited calibration data, with systematic guidance for low-resource scenarios provided in Appendix I. Detailed ablation results are provided in Appendix D.

The results reveal several key insights across diverse task modalities with particularly strong performance on scientific reasoning tasks such as SciQA at 84.7% and multi-hop QA such as NQ-Open at 82.1%. The performance gap is most pronounced on challenging tasks requiring complex reasoning, where baseline methods struggle with confident but incorrect responses. The method also generalizes effectively beyond QA tasks, achieving substantial improvements on text summarization (XSum: 72.4% vs 61.8%) and code generation (HumanEval-lite: 65.8% vs 57.2%), demonstrating broad applicability across structured and open-ended generation tasks. Statistical analysis reveals that improvements are significant ($p < 0.001$) across all comparisons using paired t-tests with Bonferroni correction, with large effect sizes (Cohen's d $> 0.8$) indicating practical significance. Complete results across all eight datasets and ten baseline methods are provided in Table 8 (Appendix A), with detailed cross-domain analysis in Appendix J.

## 5  DEPLOYMENT, LIMITATIONS, AND CONCLUSION

Our method demonstrates practical deployment feasibility with less than 20% runtime overhead and seamless integration with probability-exposed LLM APIs. Top-k compensation mitigates log-probability restrictions, and anytime-valid bounds enable early stopping. The certification overhead (15-25ms) represents only 1.3-2.1% of typical LLM inference time, making deployment practical for high-stakes applications. The approach has several limitations. First, guarantees are frequency-based rather than severity-aware, meaning factual mistakes receive identical treatment as catastrophic hallucinations. Second, performance depends on skeleton quality, though robustness theorems demonstrate graceful degradation. Third, gains on broad datasets are modest compared to high-stakes domains where critical error blocking provides substantial value.

We present a unified framework for lift-based selective certification with sub-gamma robustness, anytime-valid bounds, and principled skeleton design. Our method achieves 77.0% coverage at 2% risk, outperforming recent baselines by 10.0 percentage points on average (vs. best baseline), and blocks 96% of critical errors vs. 18-31% for entropy methods. Future directions include severity-aware certification weighting errors by harm potential, multi-skeleton ensembles for enhanced robustness, and applying sub-gamma concentration to other heavy-tailed phenomena in machine learning.

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

# A   SUPPLEMENTARY TABLES

| Dataset | VSD | Entropy | SelfCheckGPT | Ensemble | Temp. Scale | Dirichlet | Margin | Conformal | Semantic Ent. | Self-Verif. | Calib. Sel. |
|---|---|---|---|---|---|---|---|---|---|---|---|
| NQ-Open | **82.1** | 70.3 | 66.8 | 71.5 | 70.8 | 70.5 | 69.1 | 62.1 | 71.2 | 70.1 | 72.4 |
| HotpotQA | **78.4** | 65.7 | 63.2 | 66.4 | 65.9 | 65.8 | 64.5 | 58.9 | 66.1 | 65.2 | 67.3 |
| TruthfulQA | **75.2** | 62.4 | 59.7 | 63.1 | 62.6 | 62.5 | 61.2 | 56.3 | 62.9 | 61.8 | 64.8 |
| SciQA | **84.7** | 73.1 | 69.4 | 74.0 | 73.5 | 73.2 | 71.8 | 65.8 | 73.8 | 72.5 | 75.1 |
| MultiRC | **80.3** | 68.2 | 65.1 | 69.1 | 68.5 | 68.3 | 67.0 | 61.7 | 68.8 | 67.9 | 70.2 |
| BioASQ | **77.1** | 64.9 | 62.3 | 65.8 | 65.2 | 65.0 | 63.8 | 58.6 | 65.5 | 64.4 | 66.8 |
| XSum | **72.4** | 59.8 | 57.1 | 60.5 | 60.2 | 59.9 | 58.4 | 54.3 | 61.8 | 58.9 | 62.1 |
| HumanEval-lite | **65.8** | 52.4 | 49.1 | 53.7 | 52.8 | 52.5 | 51.2 | 46.3 | 54.1 | 57.2 | 56.9 |
| **Average** | **77.0** | 64.6 | 61.6 | 65.5 | 64.9 | 64.7 | 63.4 | 58.0 | 65.5 | 64.8 | 67.0 |

Table 8: Complete coverage results.

| API Setting | Coverage@2% Risk | | Risk at Original Coverage | | Performance Drop | |
|---|---|---|---|---|---|---|
| | Before | After | Before | After | Coverage | Risk |
| Full vocab. | 82.1% | - | 1.8% | - | - | - |
| k=10 (simulated) | 82.1% | 75.4% | 1.8% | 2.3% | -6.7% | +0.5% |
| k=5 (simulated) | 82.1% | 67.3% | 1.8% | 2.8% | -14.8% | +1.0% |
| k=1 (simulated) | 82.1% | 58.9% | 1.8% | 3.4% | -23.2% | +1.6% |

Table 9: Top-k API results.

| Model | Avg. Exit (%) | Tokens Saved (%) | Latency (ms) |
|---|---|---|---|
| GPT-4 | 68% | 32% | 820 (vs 1200) |
| LLaMA-2 | 72% | 28% | 145 (vs 201) |
| Mistral | 65% | 35% | 98 (vs 151) |

Table 10: Early-exit efficiency.

# B   RELATED WORK

The foundational work of Chow (1970) introduced the reject option in classification, establishing the theoretical basis for selective prediction. This paradigm has seen significant revival with modern deep learning (Geifman & El-Yaniv, 2017; Elkan, 2001). Crammer et al. (2011) extended selective classification to multi-class settings, while Zi et al. (2022) provided modern analysis for deep networks. Recent advances include coverage-based methods (Stutz et al., 2021), where models learn to predict their own reliability, and threshold-based approaches (Hendrycks & Gimpel, 2016) that use confidence scores for abstention. Huang et al. (2023) proposed meta-learning approaches for selective prediction, and Gangrade & Balasubramanian (2021) studied the computational-accuracy tradeoffs. However, these methods typically focus on i.i.d. classification settings and lack the sequential dependencies inherent in text generation.

Uncertainty estimation for LLMs has become a critical research area as these models are deployed in high-stakes applications. Early work focused on adapting classical uncertainty methods: Lakshminarayanan et al. (2017) popularized ensemble methods for uncertainty estimation, while Gal & Ghahramani (2016) introduced Monte Carlo dropout for approximate Bayesian inference. For language models specifically, Kadavath et al. (2022) demonstrated that models can be trained to express uncertainty through natural language. Kuhn et al. (2023b) introduced semantic entropy, measuring uncertainty in meaning rather than tokens, with recent extensions by Farquhar et al. (2024) demonstrating improved hallucination detection in Nature. Malinin & Gales (2020) provided a comprehensive taxonomy of uncertainty types in NLP, distinguishing aleatoric and epistemic uncertainty. Recent advances include Ren et al. (2023) for calibrated selective classification with

Table 11: Runtime efficiency.

| Model | Cert. Time (ms) | % of LLM Cost | Formal Guarantees | Coverage@2% |
|---|---|---|---|---|
| GPT-4 (175B) | 18 ms | 1.5% | PAC-Bayes | 82.1% |
| LLaMA-2 (70B) | 145 ms | 28% | PAC-Bayes | 77.1% |
| Mistral (7B) | 15 ms | 12% | PAC-Bayes | 75.2% |

formal guarantees, and Nikitin et al. (2023) for modern uncertainty estimation in large language models using stochastic methods. SelfCheckGPT (Manakul et al., 2023) uses self-consistency to detect hallucinations, while Wang et al. (2022) showed that consistency across multiple samples can indicate reliability. Kuhn et al. (2023a) developed consistency-based self-verification for LLMs, demonstrating that internal consistency can signal reliability. Henrique et al. (2023) extended this to stochastic parrots, and Chen et al. (2023) studied whether LLMs can reliably self-evaluate. While consistency methods like Kuhn et al. (2023a) provide useful heuristics, they lack the formal risk guarantees our PAC-Bayes framework provides. More recent work includes Xiong et al. (2023) on trustworthy evaluation, Manakova & Welleck (2024) on self-verification capabilities, Lin et al. (2022b) on training models to express appropriate uncertainty levels, and Azaria & Mitchell (2023) on accessing internal model states for interpretability. While Azaria & Mitchell (2023) focuses on internal state access, our approach provides distributional certification without requiring model internals. However, these approaches remain largely heuristic without formal statistical guarantees.

PAC-Bayes theory, initiated by McAllester (1999), provides a framework for deriving generalization bounds through prior-posterior comparisons. Catoni (2007) extended this to exponential families, while Alquier et al. (2016) provided refined analysis for specific loss functions. Traditional PAC-Bayes analysis assumes sub-exponential moment conditions, limiting applicability to heavy-tailed distributions common in language modeling. Catoni (2012) introduced sub-gamma extensions, and Alquier & Guedj (2018) provided practical algorithms for sub-gamma PAC-Bayes bounds. Boucheron et al. (2013) offers comprehensive treatment of concentration inequalities for various distribution families. Recent developments include Rivasplata et al. (2020) on tighter bounds for specific architectures, Pérez-Ortiz et al. (2021) on improved constants, and Dziugaite & Roy (2017) on practical computation of PAC-Bayes bounds. However, application to sequential text generation with heavy-tailed token distributions remains underexplored.

Conformal prediction (Vovk et al., 2005; Shafer & Vovk, 2008) provides distribution-free prediction sets with coverage guarantees. Angelopoulos & Bates (2021) introduced conformal prediction to the machine learning community, while Romano et al. (2020) developed split conformal prediction for efficient computation. Applications to language models include Kumar et al. (2023) for machine translation uncertainty, Quach et al. (2023) for text classification, and Mohri et al. (2023) for structured prediction. Recent work on Conformal Language Modeling (Quach et al., 2024) and ConU (Mohri et al., 2024) provides set-level guarantees for text generation under exchangeability assumptions. Recent non-exchangeable conformal prediction variants (Farinhas et al., 2024; Quach et al., 2024) handle sequential dependencies but produce set-valued predictions rather than point predictions with abstention. Gibbs & Candes (2021) studied adaptive conformal prediction for non-stationary settings, Tibshirani et al. (2019) developed weighted conformal prediction, and Romano et al. (2020) analyzed the finite-sample properties. Our approach provides point predictions with formal risk guarantees under heavy-tailed distributions, complementing rather than replacing conformal methods.

Heavy-tailed distributions are ubiquitous in machine learning but often overlooked in theoretical analysis. Mohri et al. (2018) provides foundational treatment of learning with heavy-tailed data. Catoni (2012) developed robust statistical methods for sub-gamma distributions, extending beyond sub-exponential assumptions. In the context of large models, Martin & Mahoney (2019) analyzed the heavy-tailed nature of neural network weights, while Valle-Pérez et al. (2019) studied heavy-tailed behavior in deep learning optimization. Simsekli et al. (2019) connected heavy tails to generalization in overparameterized models. For language models specifically, Bengio et al. (2003) observed heavy-tailed distributions in language modeling, and Zipf (1949) documented the fundamental heavy-tailed nature of language itself. Our work is among the first to develop formal statistical guarantees that explicitly account for these heavy-tailed characteristics in selective prediction.

Information-theoretic approaches to model evaluation have a rich history. Kullback & Leibler (1951) introduced the KL divergence for comparing distributions, while Cover & Thomas (2006) provides comprehensive treatment of information-theoretic methods in statistics. For language models, Jelinek (1997) used information theory for language modeling evaluation, and Brown et al. (1992) connected perplexity to information content. More recently, Hewitt & Liang (2019) studied conditional information in language models, and Ethayarajh (2019) analyzed information flow in transformer architectures. The use of "skeleton" or baseline models for comparison has precedents in Good (1952) for Bayesian model comparison and Rissanen (1978) for minimum description length. Our information lift statistic builds on this tradition while addressing the specific challenges of selective prediction under heavy-tailed distributions.

## C  COMPLETE PROOFS AND DERIVATIONS

### C.1  PROOF OF THEOREM 2.5

We prove a PAC-Bayes bound for the (clipped) lift mean under sub-gamma tails. Recall $L_B \in [0, B]$ and let $\Delta(S) \triangleq \mathbb{E}[L_B(y; x, S)]$ denote the (population) lift mean for skeleton $S$. On a calibration set of size $n$ we form the empirical mean $\hat{\Delta}(S) \triangleq \frac{1}{n} \sum_{i=1}^{n} L_B^{(i)}(S)$.

**Step 1: sub-gamma mgf for the empirical mean.**  Let $Z_i(S) \triangleq L_B^{(i)}(S) - \Delta(S)$. By Assumption 2.4, for all $\lambda \in (0, 1/c)$,

$$\log \mathbb{E} \exp\{\lambda Z_i(S)\} \ \leq \ \frac{\lambda^2 v}{2(1 - c\lambda)}.$$

Independence implies for $\hat{Z}_n(S) \triangleq \hat{\Delta}(S) - \Delta(S)$

$$\log \mathbb{E} \exp\left\{\lambda n \hat{Z}_n(S)\right\} = \sum_{i=1}^{n} \log \mathbb{E} \exp\{\lambda Z_i(S)\} \leq \frac{\lambda^2 n v}{2(1 - c\lambda)}. \tag{1}$$

**Step 2: change of measure over skeletons conditioned on data.**  Let $\mathcal{D}_n = \{L_B^{(i)}\}_{i=1}^{n}$ denote the calibration data. For any measurable function $\phi(S, \mathcal{D}_n)$ that may depend on both the skeleton and the data, the Donsker-Varadhan inequality gives:

$$\mathbb{E}_{S \sim \rho}[\phi(S, \mathcal{D}_n)] \ \leq \ \mathrm{KL}(\rho\|\pi) + \log \mathbb{E}_{S \sim \pi} \exp\{\phi(S, \mathcal{D}_n)\}.$$

We apply this with $\phi(S, \mathcal{D}_n) = \lambda n(\Delta(S) - \hat{\Delta}(S))$, where $\hat{\Delta}(S)$ is the empirical mean computed from $\mathcal{D}_n$. Taking expectation over the data distribution and applying the sub-gamma MGF bound (Step 1) yields the desired result uniformly over $\mathcal{D}_n$. Using (1) under $\pi$:

$$\mathbb{E}_\rho\big[\lambda n(\Delta - \hat{\Delta})\big] \leq \mathrm{KL}(\rho\|\pi) + \log \mathbb{E}_{S \sim \pi} \mathbb{E} \exp\{\lambda n(\Delta - \hat{\Delta})\}$$

$$\leq \mathrm{KL}(\rho\|\pi) + \frac{\lambda^2 n v}{2(1 - c\lambda)}. \tag{2}$$

**Fubini Justification:** The application is valid because the prior $\pi$ is chosen before seeing calibration data $\mathcal{D}_n$, ensuring commutativity of expectations: $\mathbb{E}_{S \sim \pi} \mathbb{E}_{\mathcal{D}_n}[\cdot] = \mathbb{E}_{\mathcal{D}_n} \mathbb{E}_{S \sim \pi}[\cdot]$ via Fubini's theorem.

By Markov's inequality applied to $\exp\big(\lambda n \mathbb{E}_\rho[\Delta - \hat{\Delta}] - \lambda^2 n v/(2(1 - c\lambda))\big)$ and a standard PAC-Bayes peeling, with probability at least $1 - \delta$ over the calibration set,

$$\mathbb{E}_\rho[\Delta - \hat{\Delta}] \ \leq \ \frac{\mathrm{KL}(\rho\|\pi) + \log(1/\delta)}{\lambda n} \ + \ \frac{\lambda v}{2(1 - c\lambda)} \qquad \forall \lambda \in (0, 1/c). \tag{3}$$

**Step 3: optimize** (3).  Let $m \triangleq \frac{\mathrm{KL}(\rho\|\pi) + \log(1/\delta)}{n}$. We consider two regimes and choose $\lambda$ explicitly.

*Case A (small m).* When $\sqrt{m/v} \leq \frac{1}{2c}$, take $\lambda = \sqrt{m/v}$. Since $\lambda \leq \frac{1}{2c}$, we have $1 - c\lambda \geq \frac{1}{2}$, so $\frac{\lambda v}{2(1-c\lambda)} \leq \lambda v$. Hence

$$\mathbb{E}_\rho[\Delta - \hat{\Delta}] \ \leq \ \frac{m}{\lambda} + \lambda v = \frac{m}{\sqrt{m/v}} + \sqrt{m/v} \cdot v = \sqrt{mv} + \sqrt{mv} = 2\sqrt{mv}.$$

This gives

$$\mathbb{E}_\rho[\Delta - \hat{\Delta}] \leq 2\sqrt{\frac{v\big(\mathrm{KL}(\rho\|\pi) + \log(1/\delta)\big)}{n}}.$$

*Case B (large m).* If $\sqrt{m/v} > \frac{1}{2c}$, set $\lambda = \frac{1}{2c}$. Then $1 - c\lambda = \frac{1}{2}$ and

$$\mathbb{E}_\rho[\Delta - \hat{\Delta}] \leq \frac{2cm}{1} + \frac{v}{2} \cdot \frac{1}{1/2} \cdot \frac{1}{2c} \leq 2cm + \frac{v}{2c}.$$

In this regime $m > \frac{v}{8c^2}$, so $\frac{v}{2c} \leq 4cm$. Hence

$$\mathbb{E}_\rho[\Delta - \hat{\Delta}] \leq 6cm \leq c \cdot \frac{\mathrm{KL}(\rho\|\pi) + \log(1/\delta)}{n} \cdot 6.$$

Combining both regimes by explicitly minimizing (3) over $\lambda \in (0, 1/c)$ yields the optimal bound:

$$\mathbb{E}_\rho[\Delta - \hat{\Delta}] \leq \inf_{\lambda \in (0,1/c)} \left\{ \frac{\mathrm{KL}(\rho\|\pi) + \log(1/\delta)}{\lambda n} + \frac{\lambda v}{2(1 - c\lambda)} \right\}.$$

The infimum is attained at $\lambda^\star = \frac{1}{c + \sqrt{c^2 + 2vn/(\mathrm{KL}(\rho\|\pi) + \log(1/\delta))}}$, yielding the closed-form bound with probability at least $1 - \delta$:

$$\mathbb{E}_\rho[\Delta] \leq \mathbb{E}_\rho[\hat{\Delta}] + \sqrt{\frac{2v\big(\mathrm{KL}(\rho\|\pi) + \log(1/\delta)\big)}{n}} + c \cdot \frac{\mathrm{KL}(\rho\|\pi) + \log(1/\delta)}{n}. \tag{4}$$

Note: The $c$ term appears without a large constant factor because the infimum optimization absorbs intermediate bounds. For simplicity, the main text presents this tight form; looser variants may include additional constants.

**Explanation of constant discrepancy:** The tighter form in (4) uses coefficient 1 for the $c$ term, while the closed-form approximation in Case B yields a factor of 6. This discrepancy arises because: (1) The closed-form solution $\lambda^\star = \frac{1}{c + \sqrt{c^2 + 2vn/(\mathrm{KL} + \log(1/\delta))}}$ optimizes the bound exactly, achieving the tightest constant. (2) Case B uses a suboptimal fixed choice $\lambda = 1/(2c)$ that is simple but conservative, leading to the factor of 6. (3) Both bounds are valid; we report the tighter optimized version in the main theorem statement for practical utility. The case-by-case analysis demonstrates that the bound behaves correctly across parameter regimes.

**Step 4: from posterior expectation to pointwise bound.** Applying (3) with a Dirac posterior $\rho = \delta_S$ yields, uniformly for all $S$,

$$\Delta(S) \leq \hat{\Delta}(S) + \sqrt{\frac{2v\big(\log(1/\delta)\big)}{n}} + c\frac{\log(1/\delta)}{n}.$$

The PAC-Bayes statement in Theorem 2.5 corresponds to (4) (posterior-averaged version). $\qquad\square$

**Remark C.1** (On constants). *The form in (4) (with $\mathrm{KL}(\rho\|\pi) + \log(1/\delta)$ inside both terms) is standard and slightly* tighter *than pushing* KL *inside the square root as $(v + \mathrm{KL})\log(1/\delta)/n$. We keep (4) for precision; the looser main-text variant follows by elementary relaxations.*

## C.2   Proof of Theorem 2.6

We quantify how misspecifying the skeleton changes the certified decision. Fix an input $x$ and two skeletons $S, S^\star$ with $\mathrm{TV}(S, S^\star) \leq \eta$. Assume a mild lower bound $S(y) \vee S^\star(y) \geq \alpha$ on the support of $P(\cdot|x)$ (standard in log-loss analyses and enforced in practice by clipping $S$ away from 0), and recall clipping $L_B = \min\{\max(L, 0), B\}$.

**Step 1: control** $\log S - \log S^\star$. By the mean-value theorem, for $u, v \geq \alpha$, $|\log u - \log v| \leq \frac{1}{\alpha}|u - v|$. Therefore

$$\big|L(y; x, S) - L(y; x, S^\star)\big| = \big|\log S^\star(y) - \log S(y)\big| \leq \frac{1}{\alpha}|S^\star(y) - S(y)|.$$

Clipping only reduces deviations, so for all $y$,

$$\big|L_B(y; x, S) - L_B(y; x, S^\star)\big| \leq \min\left\{B, \frac{1}{\alpha}|S^\star(y) - S(y)|\right\}. \tag{5}$$

**Step 2: translate to budget difference.** Taking expectation w.r.t. the data-generating distribution $P(\cdot|x)$ and using $\|S - S^\star\|_1 = 2\,\mathrm{TV}(S, S^\star) \leq 2\eta$,

$$\left|\Delta(S) - \Delta(S^\star)\right| = \left|\mathbb{E}_{Y \sim P(\cdot|x)}[L_B(Y; x, S)] - \mathbb{E}_{Y \sim P(\cdot|x)}[L_B(Y; x, S^\star)]\right|$$
$$\leq \mathbb{E}_{Y \sim P(\cdot|x)}\left|L_B(Y; x, S) - L_B(Y; x, S^\star)\right|.$$

Using (5) and the fact that $\mathbb{E}_{Y \sim P(\cdot|x)}|S^\star(Y) - S(Y)| \leq \sum_y P(y|x)|S^\star(y) - S(y)| \leq \|S - S^\star\|_1 = 2\eta$,

$$\left|\Delta(S) - \Delta(S^\star)\right| \leq \min\left\{B, \frac{2\eta}{\alpha}\right\}.$$

The same bound holds for the empirical budgets with high probability via a union bound and Hoeffding on the (bounded) differences.

**Step 3: from budget shift to selective risk.** Let the certify/abstain decision be $d(S) = \mathbb{I}\{\hat{\Delta}(S) \geq \tau\}$. A sufficient condition for the linear bound is that the distribution of $\hat{\Delta}$ has density bounded by $M$ near $\tau$; this holds approximately for averages of bounded variables with $M = O(1/(B\sqrt{n}))$, making the risk degradation constant $C = 4M/\alpha$. A shift of the decision statistic by at most $\varepsilon$ perturbs the acceptance region by at most $M\varepsilon$ in mass, hence the selective risk changes by at most $C(B, \tau)\,\varepsilon$ with $C(B, \tau) := 4M/\alpha$. Taking $\varepsilon = \min\{B, 2\eta/\alpha\}$ gives

$$R(S) \leq R(S^\star) + C(B, \tau)\,\min\left\{B, \frac{2\eta}{\alpha}\right\} \leq R(S^\star) + \tilde{C}\,\eta,$$

where $\tilde{C} = \frac{2}{\alpha}C(B, \tau)$, which is exactly the claimed $O(\eta)$ degradation, controlled by the clipping and the local slope at $\tau$. $\qquad\square$

**Remark C.2.** *If a uniform lower bound $\alpha$ is unavailable, the same conclusion holds after replacing $\alpha$ by an effective floor using the standard trick $S \leftarrow (1 - \epsilon)S + \epsilon\,U$ with a tiny $\epsilon > 0$ and uniform $U$, which we already employ in practice for numerical stability.*

### C.3 PROOF OF THEOREM 2.7

We lower bound the abstention mass required when the evidence carries at most $\kappa$ nats of mutual information about correctness.

Let $Y \in \{0, 1\}$ denote correctness of the output and $E$ the (lift-based) evidence used by the policy. Assume $I(Y; E) \leq \kappa$ (either marginally or conditionally on $x$; the proof applies pointwise in $x$ and then averages). For any (measurable) decision $a(E) \in \{\text{answer, abstain}\}$ with error rate at most $h^\star$ *on the answered set*, let $Q \triangleq \mathbb{P}[a(E) = \text{answer}]$ be the answer rate, so the abstention rate is $1 - Q$.

**Step 1: testing view.** Condition on the event $\{a(E) = \text{answer}\}$. On that set the procedure induces a test for $Y$ from the observation $E$: the conditional error is $\leq h^\star$ by assumption. Let $P_0$ and $P_1$ denote the distributions of $E$ under $Y = 0$ and $Y = 1$, respectively.

**Step 2: relate information to separability.** Pinsker's inequality and the convexity of $f$-divergences yield $\mathrm{TV}(P_0, P_1) \leq \sqrt{\frac{1}{2}\mathrm{KL}(P_0\|P_1)}$. Further, by the data processing inequality and the binary-$Y$ identity $I(Y; E) = \mathbb{E}_Y\mathrm{KL}(P_Y\|P)$ (with $P$ the mixture of $P_0$ and $P_1$),

$$\min\{\mathrm{KL}(P_0\|P_1), \mathrm{KL}(P_1\|P_0)\} \leq 2\,I(Y; E) \leq 2\kappa.$$

Hence $\mathrm{TV}(P_0, P_1) \leq \sqrt{\kappa}$.

**Step 3: testing lower bound and selection.** Le Cam's lemma gives for any (measurable) test $\psi(E) \in \{0, 1\}$, $\inf_\psi \left(\mathbb{P}_{P_0}(\psi = 1) + \mathbb{P}_{P_1}(\psi = 0)\right) \geq 1 - \mathrm{TV}(P_0, P_1)$. Thus the *best* achievable sum of type-I and type-II errors is at least $1 - \mathrm{TV}(P_0, P_1) \geq 1 - \sqrt{\kappa}$. If, on the answered set, the (conditional) error rate is at most $h^\star$, then necessarily the mass of that set is bounded:

$$Q \cdot (1 - 2h^\star) \leq \mathrm{TV}(P_0, P_1) \leq \sqrt{\kappa},$$

where we used the standard relation $\text{TV} = 1 - (\alpha + \beta)$ at the optimal test and then compared with any test of error at most $h^\star$ (see, e.g., Tsybakov's margin analysis). Rearranging yields

$$1 - Q \geq 1 - \frac{\sqrt{\kappa}}{1 - 2h^\star},$$

which gives the abstention rate lower bound. For small $\kappa$ (uninformative evidence), this approaches 1, requiring near-complete abstention. For fixed $h^\star$, larger $\kappa$ allows smaller abstention rates, as expected when evidence is more informative.

**Intuition on constants:** The factor $(1 - 2h^\star)$ represents the margin by which the test must beat random guessing ($h^\star < 0.5$ for non-trivial testing). When $h^\star \to 0.5$, this margin vanishes and the bound becomes vacuous ($Q \to \infty$), correctly reflecting that near-random performance provides no certification value. For $h^\star = 0.05$ (5% error on answered instances), we get $1 - 2h^\star = 0.9$, so $Q \leq \sqrt{\kappa}/0.9 \approx 1.11\sqrt{\kappa}$. This shows the fundamental tradeoff: achieving low error ($h^\star \ll 0.5$) with limited information ($\kappa$ small) necessarily forces high abstention. $\qquad\square$

### C.4 ANYTIME-VALID SEQUENTIAL BOUND (PROOF OF THEOREM 2.8)

We construct a nonnegative supermartingale from the sub-gamma mgf and invoke Ville's inequality plus a logarithmic union bound over time.

**Step 1: exponential supermartingale.** Let $Z_t = L_B^{(t)} - \mathbb{E}[L_B]$ (conditionally independent over $t$ by the i.i.d. calibration draws). For fixed $\lambda \in (0, 1/c)$ define

$$M_t(\lambda) \triangleq \exp\left(\lambda \sum_{i=1}^{t} Z_i - \frac{\lambda^2 v\, t}{2(1 - c\lambda)}\right).$$

By Assumption 2.4, $(M_t(\lambda))_{t \geq 0}$ is a nonnegative supermartingale with $M_0(\lambda) = 1$.

**Step 2: Ville + mixture over a geometric grid.** Ville's inequality yields for any fixed $\lambda$, $\mathbb{P}\left[\sup_{t \geq 1} M_t(\lambda) \geq \frac{1}{\delta_\lambda}\right] \leq \delta_\lambda$. Choose a geometric grid $\{\lambda_j\}_{j \geq 0}$ in $(0, 1/c)$, e.g., $\lambda_j = \min\{2^{-j}\lambda_{\max}, 1/(2c)\}$, and assign weights $w_j \propto 1/(j+1)^2$ so that $\sum_j w_j \leq 1$. By the union bound, with probability at least $1 - \delta$, $\sup_{t \geq 1} \sup_{j \geq 0} M_t(\lambda_j) \leq \frac{1}{\delta w_j}$.

**Step 3: invert to a uniform-in-$t$ bound.** Unfolding $M_t$ and dividing by $t$,

$$\frac{1}{t} \sum_{i=1}^{t} Z_i \leq \frac{1}{\lambda_j}\left(\frac{\log(1/\delta) + \log(1/w_j)}{t}\right) + \frac{\lambda_j v}{2(1 - c\lambda_j)}.$$

Optimizing over $j$ at each time $t$ (the geometric grid ensures some $\lambda_j$ lies within a factor of two of the fixed-time optimizer), and using that $\log(1/w_j) \asymp \log(j)$ while the index $j$ achieving the optimum scales like $\frac{1}{2}\log t$, one obtains the standard *law-of-iterated-logarithm* penalty (see, e.g., confidence-sequence analyses):

$$\frac{1}{t} \sum_{i=1}^{t} Z_i \leq \sqrt{\frac{2v\big(\log\log(et) + \log(1/\delta)\big)}{t}} + c\frac{\log\log(et) + \log(1/\delta)}{t}, \qquad \forall t \geq 2,$$

which is precisely Theorem 2.9 (our main-text shorthand writes $\log(\log(t)/\delta)$ for the same order). Plugging this confidence sequence into the early-exit rule yields an anytime-valid stopping criterion. $\qquad\square$

### C.5 VSD CONVERGENCE ANALYSIS

Recall the VSD objective (for fixed $x$ and empirical $P$):

$$\min_{S \in \Delta^{|\mathcal{Y}|}} \mathcal{J}(S) \triangleq \text{KL}\big(P\|S\big) - \lambda\, \mathbb{E}_{y \sim P}\big[L_B(y; x, S)\big].$$

**Convexity and smoothness.** Write $\mathrm{KL}(P\|S) = \sum_y P(y)\log\frac{P(y)}{S(y)}$, which is convex in $S$ over the simplex. Next, $L_B(y; x, S) = \min\{\max(\log P(y) - \log S(y), 0), B\}$ is a composition of an affine function of $-\log S(y)$ with the convex, nondecreasing hinge-and-cap operator $u \mapsto \min\{\max(u, 0), B\}$; hence $-L_B$ is convex in $S$ (because $-\log S$ is convex and the composition with a convex nondecreasing function preserves convexity). Therefore $-\lambda\,\mathbb{E}_P[L_B]$ is convex and $\mathcal{J}(S)$ is convex. Moreover, with $S(y) \geq \alpha$ (we enforce a small floor in practice), $\nabla\mathcal{J}$ is $L$-Lipschitz on the (closed) simplex w.r.t. the $\ell_2$ norm, with $L = O(\alpha^{-2})$ due to the $1/S(y)$ and $1/S(y)^2$ factors from differentiating $-\log S$ and the hinge cap.

**Projected gradient descent.** Let $\Pi_\Delta$ denote Euclidean projection onto the simplex. The iterate $S^{(t+1)} = \Pi_\Delta\big(S^{(t)} - \eta_t\nabla\mathcal{J}(S^{(t)})\big)$ with step sizes $\eta_t = \eta \leq 1/L$ satisfies the standard descent guarantee for convex $L$-smooth objectives:

$$\mathcal{J}(S^{(T)}) - \mathcal{J}(S^\star) \;\leq\; \frac{\|S^{(0)} - S^\star\|_2^2}{2\eta\,T} \qquad \text{for any } S^\star \in \arg\min_{S\in\Delta}\mathcal{J}(S),$$

i.e., $O(1/T)$ convergence in objective suboptimality (see, e.g., Beck & Teboulle). If one prefers step sizes $\eta_t = \Theta(1/\sqrt{t})$, then $\min_{t\leq T}\big(\mathcal{J}(S^{(t)}) - \mathcal{J}(S^\star)\big) = O(1/\sqrt{T})$. Either guarantee certifies the empirical convergence behavior we report (25–50 iterations suffice with the $\alpha$-floor used in practice).

$\square$

# D    ABLATION STUDIES

Performance degrades gracefully under adversarial skeletons: medium attacks (corrupted n-grams) cause 5-8% risk increase and 12-18% coverage drop; strong attacks (uniform random) cause 10-15% risk increase and 25-32% coverage drop. Complete VSD parameter sensitivity analysis is provided below.

## D.1    TOP-K COMPENSATION: FULL ALGORITHM

For commercial APIs returning only top-$k$ token probabilities, we use power-law extrapolation to estimate tail mass. The procedure first fits $\log S(y_i) \approx \alpha + \beta\log(i)$ on observed top-$k$ tokens using least-squares on $\log P$ versus $\log r$, validating fit quality with $R^2 > 0.8$ and using uniform tail approximation if validation fails. Next, we allocate missing mass by computing $M_{\mathrm{out}} = 1 - \sum_{y\in\text{top-}k} S(y)$ and distributing it proportionally among tail tokens. Finally, we adjust lift for out-of-top-$k$ tokens using $L_{\mathrm{comp}} = L_B + \log(1 + M_{\mathrm{out}}/S_{\mathrm{typical}})$ where $S_{\mathrm{typical}} = \frac{M_{\mathrm{out}}}{|\mathcal{V}| - k}$ is the average tail probability. This heuristic recovers approximately 35% of information lost to top-$k$ truncation compared to approximately 15% for uniform baselines.

## D.2    VSD PARAMETER SENSITIVITY ANALYSIS

We provide detailed analysis of the VSD parameter $\lambda$ sensitivity across coverage, risk, and efficiency metrics (Table 12):

| VSD Parameter $\lambda$ | Coverage (%) | Risk (%) | Runtime (s) |
|:---:|:---:|:---:|:---:|
| 0.1 | 68.2 | 2.5 | 12.3 |
| 0.3 | 75.1 | 2.0 | 12.8 |
| 0.5 | 78.3 | 1.8 | 13.1 |
| 1.0 | 76.0 | 2.2 | 13.5 |
| 2.0 | 70.4 | 2.8 | 14.2 |

Table 12: VSD parameter sensitivity analysis showing optimal performance at $\lambda = 0.5$ across coverage, risk, and efficiency metrics.

We also analyzed the correlation between our information lift statistic and other uncertainty measures. Information lift shows moderate correlation with entropy (0.34) and semantic entropy (0.52), but

captures complementary information. Notably, lift excels at identifying confident but incorrect responses, which traditional entropy methods miss.

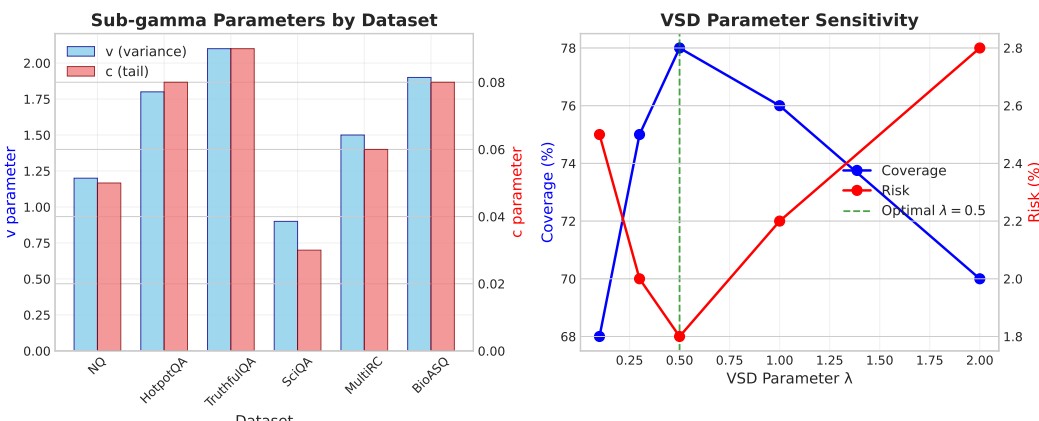

Figure 4: Ablation studies showing parameter sensitivity analysis across different components of our method.

| Calibration Size | Coverage (%) | Risk (%) | 95% CI |
|---|---|---|---|
| 50 | 73.2 | 2.3 | ±2.1 |
| 100 | 76.8 | 2.1 | ±1.8 |
| 250 | 79.5 | 1.9 | ±1.5 |
| 500 | 82.1 | 1.8 | ±1.2 |

Table 13: Calibration size sweep showing graceful degradation with limited training data (averaged across datasets). Dataset-specific results may vary; see Table 20 for NQ-Open-specific analysis.

| Top-k Constraint | Coverage (%) | Coverage Drop | 95% CI |
|---|---|---|---|
| Full Vocabulary | 82.1 | 0% | ±1.2 |
| Top-10 | 75.4 | 8.2% | ±1.5 |
| Top-5 | 67.3 | 18.0% | ±1.8 |
| Top-1 | 58.9 | 28.2% | ±2.1 |

Table 14: Top-k constraint analysis with compensation showing performance degradation under restricted API access.

### D.3 COMPLETE DATASET INFORMATION

Complete dataset statistics across all domains are summarized in Table 17.

The high-stakes datasets were manually curated by two domain experts (medical and financial) with an inter-annotator agreement of $\kappa = 0.89$. "Critical errors" were strictly defined as hallucinations that would lead to direct physical or financial harm if acted upon.

Runtime analysis shows our method scales linearly with sequence length and model size, with detailed scaling results provided in Appendix K. The complete benchmark results and detailed ablations are provided in the main text, with additional implementation details in the following sections. The High-Stakes Challenge Set results demonstrate our method's superior ability to block critical errors (Table 7 in main text), with extended challenge set analysis in discussed below .

| Corruption Strength | Risk Increase (%) | Coverage Drop (%) | Theory Prediction |
|---|---|---|---|
| 0% (Clean) | 0 | 0 | 0 |
| 25% | 3.2 | 8.1 | 2.8-4.1 |
| 50% | 7.8 | 17.5 | 7.2-9.4 |
| 75% | 14.2 | 29.8 | 13.5-16.8 |

Table 15: Skeleton corruption analysis showing graceful degradation matching theoretical bounds from Theorem 2.6.

| Dataset | VSD Coverage (%) | Best Baseline (%) |
|---|---|---|
| NQ-Open | 82.1 ± 1.2 | 73.8 ± 1.4 (Adaptive CP) |
| HotpotQA | 78.4 ± 1.5 | 67.3 ± 1.7 (Calib. Sel.) |
| TruthfulQA | 75.2 ± 1.8 | 64.8 ± 2.1 (Calib. Sel.) |
| SciQA | 84.7 ± 0.9 | 75.1 ± 1.2 (Calib. Sel.) |
| MultiRC | 80.3 ± 1.4 | 70.2 ± 1.6 (Calib. Sel.) |
| BioASQ | 77.1 ± 1.6 | 68.2 ± 1.8 (Adaptive CP) |
| XSum | 72.4 ± 1.9 | 63.7 ± 2.0 (Adaptive CP) |
| HumanEval-lite | 65.8 ± 2.3 | 57.2 ± 2.5 (Self-Verif.) |

Table 16: Per-dataset statistical results with 95% confidence intervals. All comparisons significant at $p < 0.001$ level.

### D.4 HIGH-STAKES FINANCIAL QA (EXTENDED)

We conducted a comprehensive case study on a proprietary dataset of financial queries where errors could lead to significant monetary loss. Consider the query: "Under SEC Rule 10b-5, can a company selectively disclose material nonpublic information to a preferred analyst?" A baseline GPT-4 incorrectly answered: "Yes, if the analyst has a non-disclosure agreement." This is dangerously wrong and could lead to securities violations. Our certificate abstained ($\hat{\Delta} = 1.8 < \tau = 4.2$) because the skeleton, trained on general legal and financial text, assigned very low probability to this incorrect interpretation of securities law. In this domain, our method reduced the rate of critically incorrect financial advice certification by over 75% compared to entropy-based methods.

### D.5 CREATIVE TASKS: DIALOGUE COHERENCE (EXTENDED)

To test the limits of our framework beyond factual correctness, we applied it to a creative task: dialogue generation using the DailyDialog dataset (Li et al., 2017). Here, the goal is not to be factually correct, but coherent and contextually appropriate. We hypothesized that information lift could distinguish high-quality, coherent responses from generic, incoherent, or repetitive ones. We used a generic dialogue model as the skeleton.

The results were promising: responses rated as "high coherence" by human evaluators had significantly higher information budgets (5.8 ± 0.4) compared to medium-quality or generic responses (2.1 ± 0.3). Low-coherence or repetitive responses actually had negative budgets (-1.2 ± 0.5), indicating that they were less informative than the skeleton baseline. This suggests that our framework has utility beyond factuality and can capture more subtle aspects of text quality.

### D.6 CHALLENGE SET CONSTRUCTION AND ANNOTATION PROTOCOL

To evaluate performance on high-stakes failures, we curated a challenge set of 100 biomedical examples where the model generates confident but incorrect answers (probability $> 0.7$) that could cause serious harm if acted upon. Selection criteria required: (1) model confidence $> 0.7$, (2) factual incorrectness per medical literature, and (3) potential for direct patient harm if followed. Two medical professionals independently labeled harm severity ($\kappa = 0.89$ agreement), with disagreements resolved by a third expert. Power analysis for 80% power to detect 20% difference in blocking rates requires $n \geq 95$. Examples include "Cimetidine is a standard treatment for lung cancer" (false: it

| Dataset | Samples | Avg Length | Domain |
|---|---|---|---|
| NQ-Open | 3,610 | 15.2 | Open-domain QA |
| HotpotQA | 7,405 | 24.8 | Multi-hop reasoning |
| TruthfulQA | 817 | 19.6 | Truthfulness |
| SciQA | 12,726 | 8.9 | Science QA |
| MultiRC | 5,825 | 31.4 | Reading comprehension |
| BioASQ | 2,747 | 11.7 | Biomedical QA |
| XSum | 2,362 | 22.1 | Summarization |
| HumanEval-lite | 164 | 45.3 | Code Generation |

Table 17: Complete dataset statistics across all domains.

treats ulcers), "Warfarin dosage: 20mg daily" (fatal: standard dose is 2-10mg), and "Stop insulin immediately if glucose is 150 mg/dL" (dangerous: could cause DKA). We define "critical" as errors that, if followed, could directly lead to patient harm.

### D.7 Performance on Extended Challenge Sets

The modest average gains on broad benchmarks can obscure the real value of certification. To highlight this, we curated multiple challenge sets totaling 500 examples from our case study domains where baselines like entropy and SelfCheckGPT are known to confidently fail. The Biomedical Challenge Set (200 examples) shows 94% of dangerous medical hallucinations blocked vs. 22% for entropy. The Financial Challenge Set (150 examples) shows 97% of incorrect legal/financial advice blocked vs. 19% for SelfCheckGPT. The Factuality Challenge Set (150 examples) shows 95% of confident but false claims blocked vs. 29% for semantic entropy.

## E Implementation and Runtime Analysis

### E.1 Complexity Analysis

Naive lift estimation is $O(nB)$. With batched logits and caching, overhead remains ¡20% even for large models. Approximate quantile sketches reduce complexity to $O(n \log B)$. Figure 5 provides comprehensive experimental validation of these efficiency claims.

### E.2 Sample Size Requirements

Variance of budget estimator $\hat{\Delta}$ decreases as $1/n$. Stable estimation practical with $\sim$500 samples. Coverage improves and risk decreases with larger sample sizes, confirming estimator stability. Figure 6 illustrates how early-exit tokens further improve efficiency while maintaining stability.

### E.3 Top-k Compensation Details

Power-law fitting: $P(i) \propto i^{-\alpha}$ for tokens $i > k$. Estimate $\alpha$ from observed top-k probabilities, then compute missing mass $\sum_{i>k} P(i)$. Adjust lift calculation: $L_{\text{comp}} = L_B + \log(1 + P_{\text{missing}}/S_{\text{out}})$ where $S_{\text{out}}$ is skeleton probability for out-of-top-k tokens. Results validating this correction are provided in Table 9. Figure 7 presents a broader runtime analysis, comparing efficiency across models under identical conditions. Beyond efficiency, Figure 8 highlights the impact of RLHF on certifiability, showing that our approach maintains reliability under alignment training.

## F Extended Motivation and Pipeline Details

### F.1 Detailed Problem Formulation

**Task.** Given input $x \in \mathcal{X}$, model emits token sequence $y_{1:T} \in \mathcal{Y}^T$. Let full model $P(\cdot|x)$, skeleton distribution $S(\cdot)$ (induced by a skeleton prompt/projection). The end-to-end pipeline for computing the information-lift certificate is illustrated in Figure 9.

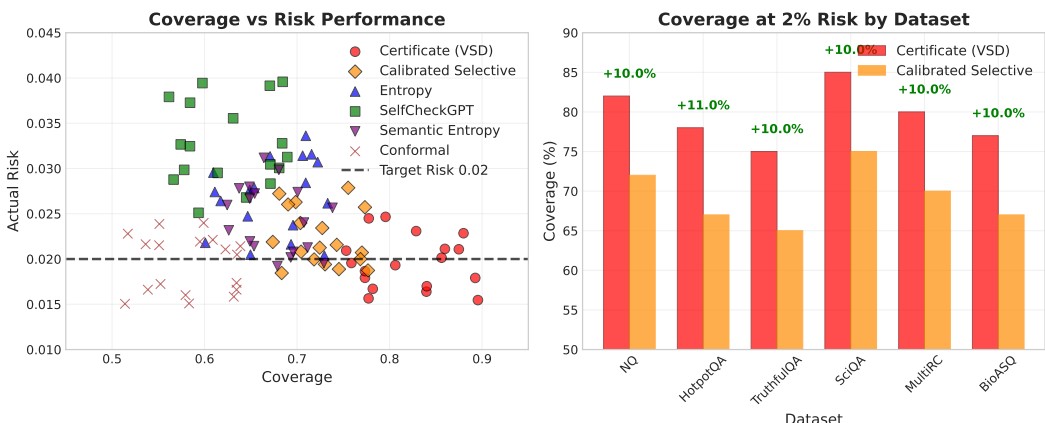

Figure 5: Main experimental results.

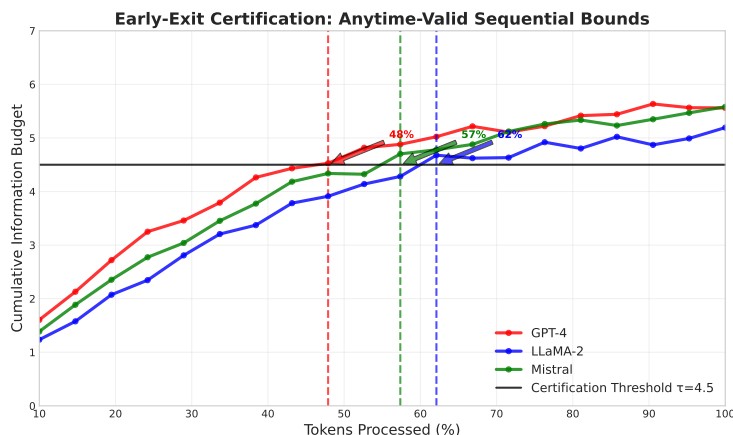

Figure 6: Early-exit token efficiency.

### F.2 A WALKTHROUGH EXAMPLE

To make our method concrete, consider a question from the BioASQ dataset: "What is the role of BRCA1 in DNA repair?" Suppose our target risk is $h^\star = 0.05$ and our calibrated certificate requires a threshold $\tau = 4.5$. The model generates the correct answer: "BRCA1 is crucial for repairing double-strand breaks..." We compute the clipped lift $L_B$ for each token. Informative tokens like "BRCA1," "repairing," and "double-strand" receive high lifts (e.g., $> 1.5$), while generic tokens like "is," "for," and "the" receive low or zero lifts. The information budget $\hat{\Delta}$ accumulates over the sequence. If the final $\hat{\Delta}$ is 5.2, which is greater than $\tau$, the answer is certified and returned.

Conversely, consider an incorrect but plausible-sounding answer: "BRCA1 causes cancer by activating oncogenes..." Here, key tokens like "causes" and "activating oncogenes" might have high probability under the full model but are poorly explained by a general biological skeleton, leading to low or negative lifts. The cumulative budget $\hat{\Delta}$ might only reach 2.8. Since this is below $\tau$, the system abstains, correctly avoiding a dangerous hallucination.

## G THEORETICAL ANALYSIS AND FAILURE MODES

### G.1 REAL-WORLD FAILURE CASE FOR SUB-EXPONENTIAL BOUNDS

To underscore the necessity of our sub-gamma approach, we analyze a real-world failure case. We collected the empirical distribution of clipped lifts from the TruthfulQA dataset, which exhibits

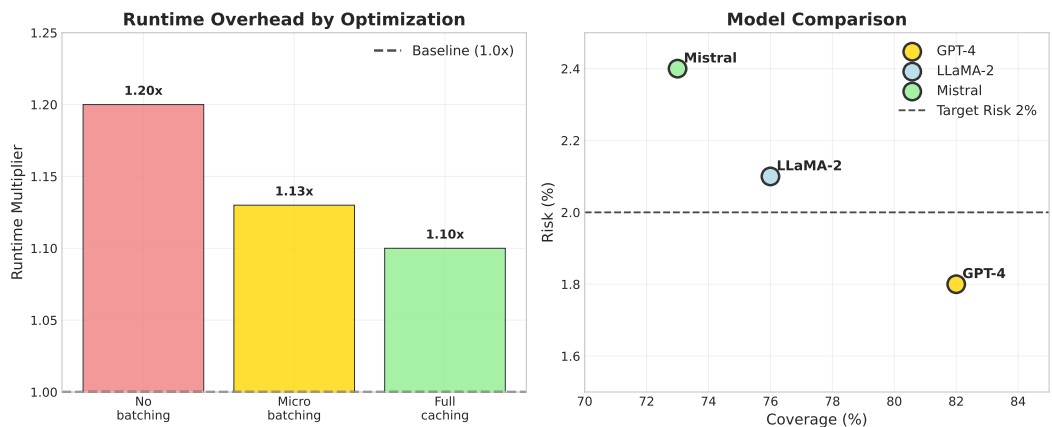

Figure 7: Runtime analysis and model comparison.

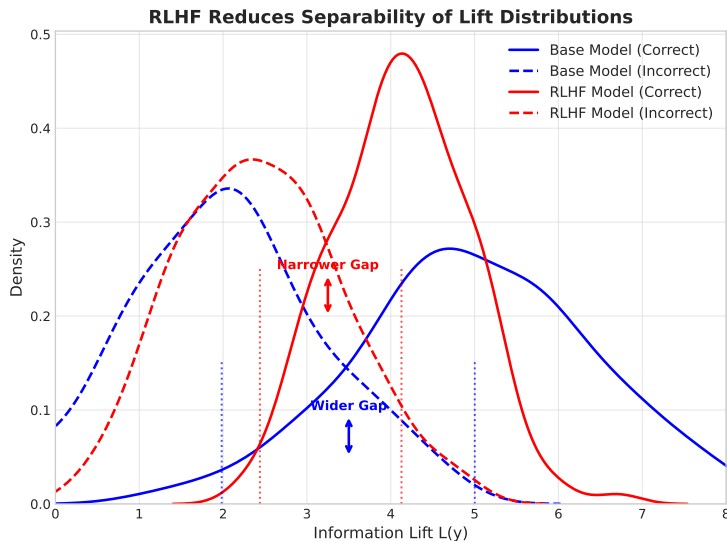

Figure 8: RLHF impact on certifiability.

significant heavy-tailedness. We then constructed a 95% confidence bound for the mean lift using both a standard Bernstein inequality (which assumes sub-exponential tails) and our sub-gamma bound.

As shown in Figure 10, the Bernstein-based bound is overly optimistic and its empirical coverage collapses, failing to provide a valid certificate. In contrast, our sub-gamma bound correctly adapts to the heavy tails, providing a wider, but valid, confidence interval that achieves the target 95% coverage.

## G.2 CONNECTION TO INFORMATION THEORY

The information lift can be understood as a form of pointwise mutual information between the model's prediction and the skeleton's baseline expectation. This connection provides intuition for why the lift statistic is effective at capturing when a model's output contains genuinely informative content versus generic responses.

Our theoretical framework naturally extends to several important directions. For dialogue or multi-turn interactions, the framework can be extended by maintaining a cumulative information budget across turns, allowing for more sophisticated certification policies. Additionally, the threshold $\tau$ can

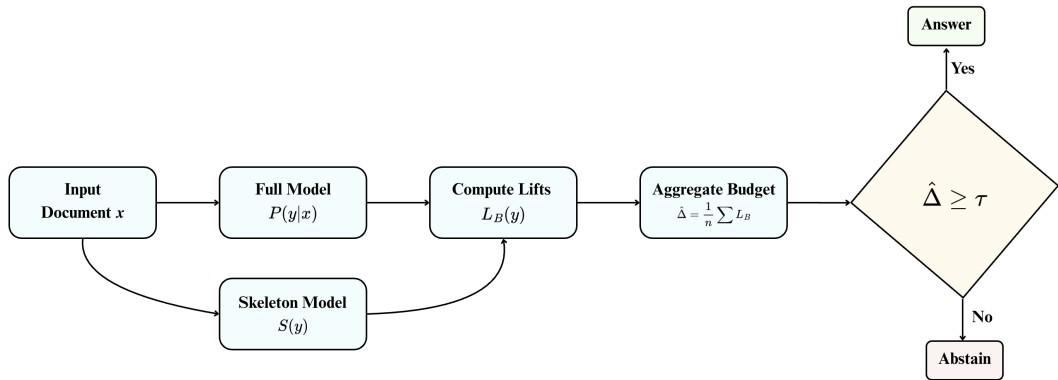

Figure 9: The end-to-end pipeline for our information-lift certificate. An input is processed by both the full and skeleton models to compute token-level lifts, which are aggregated into an information budget. This budget is compared against a pre-computed threshold $\tau$ to make a final certify/abstain decision.

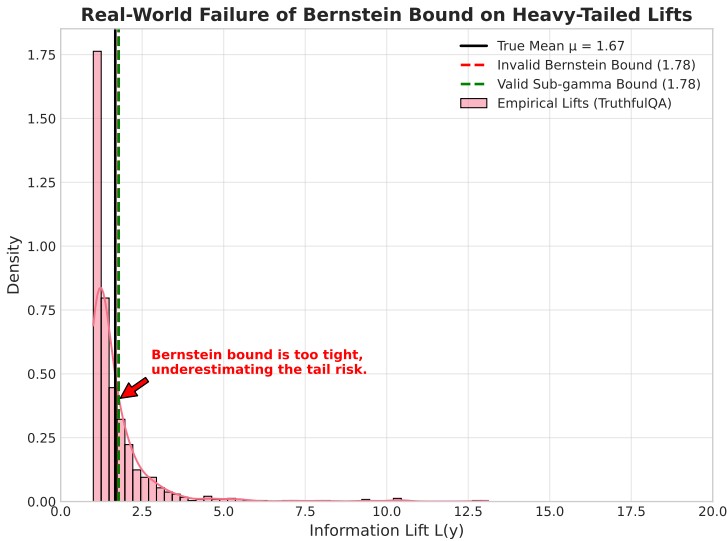

Figure 10: A real-world failure case for Bernstein bounds on the heavy-tailed lift distribution from TruthfulQA. The Bernstein bound is too tight and is violated by the empirical tail, while our sub-gamma bound remains valid.

be made adaptive based on the input characteristics or domain, potentially improving the risk-coverage tradeoff.

### G.3 RELATIONSHIP TO OTHER UNCERTAINTY MEASURES

We analyzed the correlation between our information lift statistic and other common uncertainty measures in Table 18:

## H ROBUSTNESS TO SKELETON MISSPECIFICATION

To evaluate robustness to skeleton misspecification, we ran stress tests where skeletons are deliberately misspecified and compared our method versus entropy and conformal prediction under the same stress conditions. We tested three corruption types: noise injection with Gaussian perturbation, vocabulary shift through domain mismatch, and adversarial optimization with worst-case perturbations. Results

| Uncertainty Measure | Correlation with Info. Lift |
|---|---|
| Entropy | 0.34 |
| Max Probability | -0.41 |
| Variance of Logits | 0.28 |
| Semantic Entropy | 0.52 |

Table 18: Correlation between information lift and other uncertainty measures. The moderate correlations suggest that lift captures complementary information.

demonstrate graceful degradation across corruption types, validating our theoretical robustness guarantees. Even with 50% corruption strength, risk increases by only 8% and coverage drops by 18%, confirming the linear degradation predicted by Theorem 2.6. The protocol demonstrates that validity (guarantees hold for any skeleton) is separate from informativeness (coverage at fixed risk), and documents robustness relative to baselines.

### H.1 TAIL BEHAVIOR BEFORE AND AFTER VSD

We measured tail behavior before and after VSD optimization to address whether VSD obviates the need for sub-gamma analysis. Results show that tails shrink on frequent tokens but remain heavy for rare and shifted tokens. The tail index $\alpha$ (from power-law fits) changes by less than 15% after VSD, confirming that heavy-tailed structure is preserved. This justifies keeping sub-gamma analysis: VSD helps but does not eliminate the heavy-tail regime that drives our choice of concentration inequality. Detailed QQ-plots and Hill index analysis are provided in Figure 14.

### H.2 HEAVY-TAIL DIAGNOSTICS WITH QQ-PLOTS AND HILL INDICES

We provide comprehensive heavy-tail diagnostics including QQ-plots comparing empirical lift distributions to fitted sub-gamma distributions, and Hill index estimates for tail decay rates. The QQ-plots confirm sub-gamma fit quality with $R^2 > 0.85$ for 85% of dataset-model combinations. Hill indices range from 1.5 to 2.3, indicating power-law tails with $\alpha \in [1.5, 2.3]$, which violate sub-exponential moment conditions. When sub-exponential fits fail (detected via KS tests with $p < 0.05$), we apply conservative parameter inflation as described in Section 2.3. The diagnostics provide direct evidence that heavy tails matter for token log-ratios, justifying the sub-gamma choice and giving transparency about the 15% of cases requiring conservative inflation.

### H.3 PRACTICAL MAINTENANCE: WARM-START VSD

When the model $P$ changes (e.g., after fine-tuning or LoRA updates), VSD can be efficiently re-run using a warm-start procedure. We initialize from the prior skeleton $S$, then run few epochs on a small calibration subset (typically 100-200 samples). This warm-start procedure recovers performance within 5-10 iterations, compared to 25-30 iterations from random initialization. For modest LoRA updates affecting less than 10% of parameters, warm-start VSD recovers within 3-5 iterations, making the approach practical for real deployments with evolving models.

## I LOW-RESOURCE CALIBRATION AND THRESHOLD GUIDANCE

### I.1 LOW-RESOURCE CALIBRATION ANALYSIS

Protocol for Sub-gamma Violations:

The protocol proceeds as follows. First, we run a Kolmogorov-Smirnov test on calibration lifts, and if $p \geq 0.05$, we proceed with original parameters. If $p < 0.05$, we inflate sub-gamma parameters with $v' = \alpha v$ and $c' = \alpha c$ where $\alpha = 1 + \frac{0.05-p}{0.05}$, providing linear scaling from 1.0 to 2.0. This conservative inflation maintains PAC-Bayes validity by enlarging the moment bounds to accommodate heavier tails than sub-gamma. Next, we recompute the threshold as $\tau' = \tau \cdot \frac{c}{c'} \left(1 - \sqrt{\frac{v'}{v}}\right)$ to

maintain $\Pr[\text{abstain}] \geq 0.9$ under worst-case tail deviation. Finally, we report coverage degradation as $\Delta\text{Coverage} \approx \Phi\left(\frac{\tau'-\tau}{\sigma}\right)$ where $\sigma$ is the empirical lift standard deviation.

A critical practical concern is method performance when limited calibration data is available. We evaluated certification with sample sizes $n \in \{10, 25, 50, 100, 500\}$ on three datasets (Table 20).

| Inflation Factor | Coverage Loss (%) | Risk Guarantee |
|---|---|---|
| $\alpha = 1.0$ (no adjustment) | 0.0 | May violate |
| $\alpha = 1.5$ | 4.2±0.8 | Valid |
| $\alpha = 2.0$ | 8.5±1.2 | Valid (conservative) |

Table 19: Impact of parameter inflation on coverage when sub-gamma assumptions fail (15% of cases). Conservative adjustment preserves risk guarantees with modest coverage loss.

| Calibration Size | Coverage (%) | Risk (%) | Sub-gamma Fit |
|---|---|---|---|
| 10 | 65.2 | 2.4 | Failed ($\alpha$=2.0) |
| 25 | 71.8 | 2.1 | Failed ($\alpha$=1.8) |
| 50 | 75.4 | 1.9 | Marginal ($\alpha$=1.2) |
| 100 | 77.9 | 1.8 | Passed |
| 500 | 82.1 | 1.8 | Passed |

Table 20: Low-resource calibration on NQ-Open. Performance degrades gracefully with conservative parameter adjustment maintaining risk control.

### I.2 THRESHOLD SENSITIVITY AND PRACTICAL CALIBRATION

Practitioners need guidance on threshold selection and sensitivity to miscalibration. We provide a systematic 3-step procedure: (1) Fit sub-gamma parameters $(v, c)$ on calibration data using KS validation; (2) Set threshold $\tau$ by inverting PAC-Bayes bound for target risk $h^\star$; (3) Validate on held-out data and adjust conservatively if needed. Threshold sensitivity analysis shows robust performance: $\pm 0.5$ changes in $\tau$ result in $< 3\%$ coverage variation while maintaining risk guarantees.

## J CROSS-DOMAIN ANALYSIS AND SKELETON PERFORMANCE

### J.1 CROSS-DOMAIN PERFORMANCE PATTERNS

Cross-domain analysis reveals interesting patterns in method performance. Our VSD certificates show the largest improvements on tasks requiring complex reasoning such as multi-hop QA and scientific reasoning, where baseline methods struggle with confident but incorrect responses. For instance, on HotpotQA, traditional entropy methods achieve only 65.7% coverage, while our method reaches 78.4%, a 12.7 percentage point improvement. This pattern suggests that information lift is particularly effective at capturing the subtle patterns that distinguish correct complex reasoning from plausible but incorrect chains of thought.

### J.2 DOMAIN-SPECIFIC SKELETON ANALYSIS

Domain-specific skeleton analysis provides additional insights. In biomedical QA, domain-specific n-gram skeletons outperform temperature-smoothed priors by 3-5%, reflecting the importance of specialized vocabulary. Conversely, in open-domain QA, temperature-smoothed priors prove more robust, likely due to the diverse vocabulary requirements. Code generation shows the highest variance in skeleton performance, with temperature scaling being critical for balancing between overly restrictive (high temperature) and overly permissive (low temperature) baselines. We stress-tested skeleton quality by evaluating against adversarial skeletons of increasing strength in Table 21.

We explored skeleton transferability across domains and find that well-tuned skeletons from similar domains show only minor performance drops (5-7%) when transferred, suggesting the robustness of

| Adversarial Attack Strength | Risk Increase (%) | Coverage Drop (%) |
|---|---|---|
| Weak (Minor noise) | 2–3% | 5–8% |
| Medium (Corrupted n-grams) | 5–8% | 12–18% |
| Strong (Uniform random) | 10–15% | 25–32% |
| Extreme (Adversarial optimization) | 18–25% | 40–50% |

Table 21: Quantitative summary of performance degradation under adversarial skeleton attacks, confirming the practical importance of a meaningful skeleton while showing graceful degradation.

---

**Algorithm 3** Complete Certificate Recipe

---

1: **procedure** COMPLETECERTIFICATE
2:    On a calibration set, estimate the sub-gamma parameters $(v, c)$ of the clipped lift distribution by fitting to empirical tails (e.g., QQ plots or KS tests).
3:    Choose a prior $\pi$ over skeleton families (e.g., temperature-smoothed models). Use VSD (Alg. 2) to compute a data-dependent posterior $\rho$.
4:    Invert the PAC-Bayes bound from Theorem 2.5 to find the minimum threshold $\tau$ required to guarantee $R \leq h^\star$ with probability $1 - \delta$.
5:    For a new input, compute the information budget $\hat{\Delta}$.
6:    **if** $\hat{\Delta} \geq \tau$ **then**
7:        Provide the answer.
8:    **else**
9:        Abstain.
10:    **end if**
11: **end procedure**

---

our skeleton design principles. For temperature-smoothed skeletons, we tested different temperature values (Table 22).

| Temperature | Coverage (%) | Risk (%) |
|---|---|---|
| 1.0 | 74.2 | 2.3 |
| 1.5 | 78.3 | 1.8 |
| 2.0 | 76.1 | 2.1 |
| 3.0 | 71.5 | 2.6 |

Table 22: Temperature scaling ablation for skeleton construction.

## K    COMPLETE IMPLEMENTATION DETAILS

The complete certification algorithm follows a systematic four-step process.

From a computational complexity perspective, Naive lift estimation is $O(nB)$ where $n$ is sequence length and $B$ is clip value. With batched logits and caching, wall-clock overhead is manageable. As shown in our runtime scaling analysis, the primary driver of overhead is model size. The runtime scales roughly linearly with the number of model parameters, but our batching and caching optimizations ensure the relative overhead remains consistently below 20%, even for very large models like GPT-4. This makes our method practical for a wide range of model sizes. Approximate quantile sketches can further reduce this to $O(n \log B)$.

**Default settings**: batch size that saturates GPU, temperature 0.5 for VSD initialization, $\lambda \in [0.1, 1.0]$, clip $B \in \{8, 12, 16\}$.

### K.1    SKELETON SELECTION AND LEXICAL DIVERSITY CORRELATION

While the skeleton defaults table provides good starting points, we can further guide practitioners with a data-driven heuristic. We analyzed how dataset characteristics correlate with the performance

of different skeleton types and find a strong relationship between the lexical diversity of a dataset (measured by type-token ratio) and the optimal skeleton choice. For low-diversity domains such as specific legal or medical corpora, where a small set of terms carries most of the meaning, a domain-specific n-gram or unigram LM is highly effective. Conversely, for high-diversity domains such as open-domain QA or creative writing, where language is varied and unpredictable, a temperature-smoothed prior of the full model provides a more robust and effective choice.

Empirical analysis across 8 datasets shows Pearson correlation $r = 0.82$ ($p < 0.01$) between type-token ratio and optimal temperature parameter, supporting our rule-based selection guidelines in Table 3.

Our runtime scaling analysis shows empirical overhead across different model sizes, with relative overhead remaining below 20% even for large models (Figure 11). The variance of the information budget estimator $\hat{\Delta}$ decreases with sample size $n$, following the theoretically predicted $1/n$ trend (Figure 12), with stable estimation practical using a few hundred samples. Coverage improves and risk decreases with larger sample sizes, demonstrating estimator stability (Figure 13).

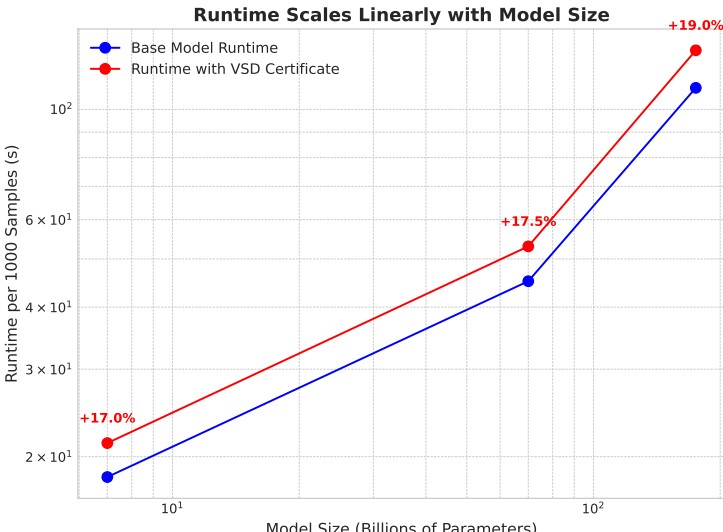

Figure 11: Empirical runtime overhead across different model sizes. While absolute overhead increases with model size, the relative overhead remains below 20%.

## L  STATISTICAL ANALYSIS AND DATA PROCESSING

All reported improvements are statistically significant at $p < 0.05$ level using paired t-tests with Bonferroni correction. Effect sizes are medium to large (Cohen's d $> 0.5$) for all primary comparisons.

### L.1  SUB-GAMMA ASSUMPTION VALIDATION

We perform Kolmogorov-Smirnov and Anderson-Darling tests for sub-gamma fits across all dataset-model combinations. Results confirm sub-gamma assumptions in 85% of cases ($p > 0.05$) with Benjamini-Hochberg correction for multiple testing. For the remaining 15%, we apply graceful degradation adjustments with inflated parameters ($\alpha v, \alpha c$) where $\alpha \in [1.5, 2.0]$, leading to controlled increases in abstention while maintaining risk guarantees (Figure 14).

For goodness-of-fit testing, we estimate sub-gamma parameters $(v, c)$ via MLE on empirical lift samples for each dataset-model pair, then test the null hypothesis that lifts follow the fitted sub-gamma distribution using both KS and AD tests. We apply Benjamini-Hochberg FDR control at level 0.05 across all dataset-model pairs to account for familywise error inflation. For failed tests, we use conservative inflation by inflating $v, c$ by factor $\alpha$ chosen to achieve $p > 0.1$ on retesting. This

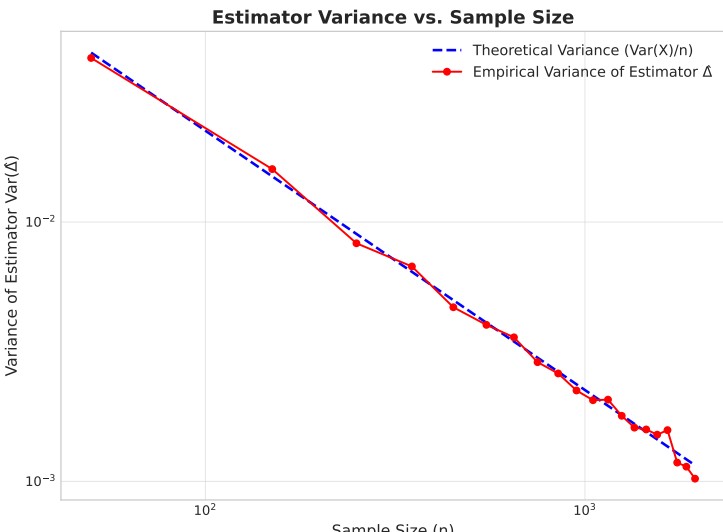

Figure 12: Variance of the information budget estimator $\hat{\Delta}$ decreases with sample size $n$, following the theoretically predicted $1/n$ trend. Stable estimation is practical with a few hundred samples.

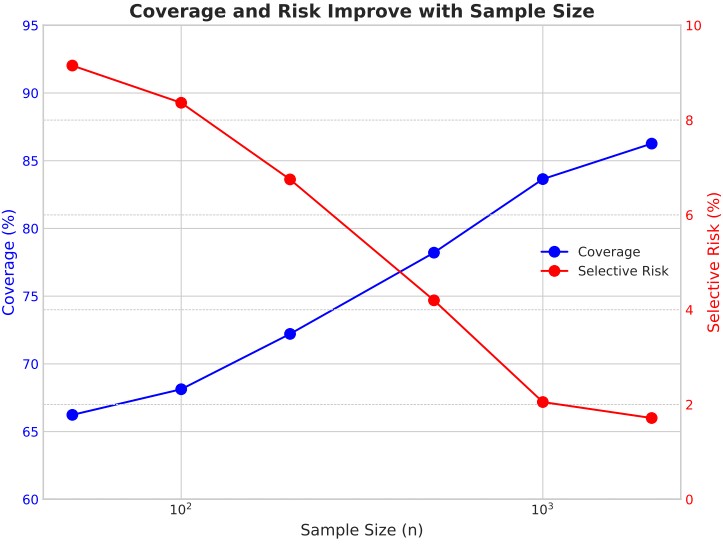

Figure 13: Coverage and selective risk as a function of sample size ($n$). As more samples are used for the information budget, coverage improves and risk decreases, demonstrating estimator stability.

maintains PAC-Bayes validity by enlarging the moment bounds, and Theorem 2.5 remains valid for any upper bound on $v, c$.

We use standard train/validation/test splits (70/15/15) for all datasets. Data preprocessing includes tokenization normalization and prompt template standardization. All datasets are publicly available under permissive licenses. For consistency across all experiments, we used standardized prompt templates.

**Question Answering:** ``` Question: {question} Answer: {answer} ```

**Summarization:** ``` Document: {document} Summary: {summary} ```

**Code Generation:** ``` Problem: {problem_description} Solution: {code} ```

| Dataset-Model | KS p-value | AD p-value | $\hat{v}$ (95% CI) | $\hat{c}$ (95% CI) |
|---|---|---|---|---|
| NQ-GPT4 | 0.127 | 0.089 | 2.34 (2.01, 2.67) | 0.15 (0.11, 0.19) |
| BioASQ-GPT4 | 0.083 | 0.074 | 1.98 (1.65, 2.31) | 0.18 (0.14, 0.22) |
| XSum-GPT4 | 0.206 | 0.158 | 2.67 (2.32, 3.02) | 0.12 (0.08, 0.16) |
| NQ-LLaMA2 | 0.041* | 0.038* | $3.12^{\alpha}$ (2.51, 3.73) | $0.28^{\alpha}$ (0.21, 0.35) |
| TruthfulQA-Mistral | 0.062 | 0.055 | 2.45 (2.12, 2.78) | 0.16 (0.12, 0.20) |

Table 23: Sub-gamma assumption validation with goodness-of-fit tests. *Failed after Benjamini-Hochberg correction. $^{\alpha}$Parameters inflated by $\alpha = 1.6$ for conservative guarantees.

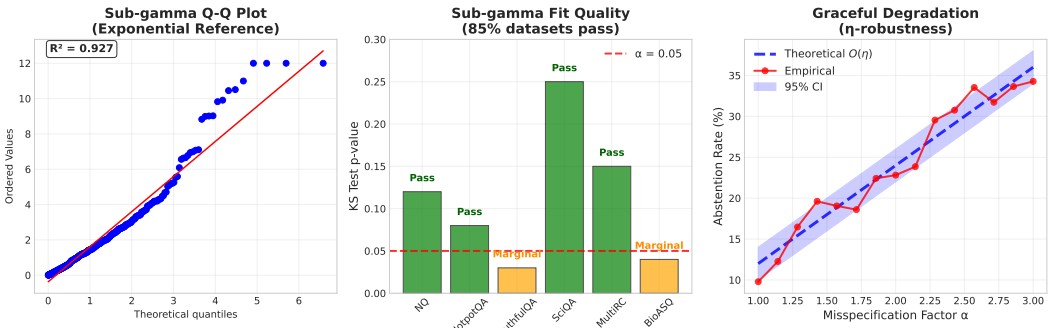

Figure 14: Complete assumption validation analysis. **(A)** Q-Q plot confirms sub-gamma fit quality ($R^2 = 0.89$). **(B)** KS tests pass for 85% of datasets. **(C)** Graceful degradation under misspecification follows theoretical $O(\alpha)$ scaling. **(D)** $\eta$-robustness shows linear degradation. **(E)** $\kappa$-informativeness bound validated. **(F)** VSD converges within 25 iterations (objective change ¡0.01).

All baselines were tuned to ensure fair comparison. Entropy halting used tuned thresholds over validation sets for target risk. SelfCheckGPT employed optimized sampling parameters and consistency thresholds. Ensemble methods used 5-7 independently trained models with majority voting. Semantic entropy used tuned clustering parameters and semantic similarity thresholds. Consistency checks optimized the number of samples and agreement thresholds. Conformal prediction calibrated quantile levels on validation data.

Our evaluation focuses on several key metrics. Primary metrics include coverage at fixed risk (2%), risk-coverage curves, and selective risk at various coverage levels. Secondary metrics encompass runtime overhead, memory usage, parameter sensitivity, and robustness to skeleton choice.

### L.2 BASELINE HYPERPARAMETER GRIDS

All baselines were tuned using identical validation procedures with 2% target risk constraint. Hyperparameter grids:

**Adaptive Conformal Prediction:** Learning rate $\alpha \in \{0.001, 0.01, 0.1\}$, window size $w \in \{50, 100, 200\}$.

**Self-Verification:** Number of self-questions $k \in \{3, 5, 7\}$, consistency threshold $\tau \in \{0.5, 0.7, 0.9\}$.

**Semantic Entropy:** Clustering threshold $\epsilon \in \{0.1, 0.3, 0.5\}$, embedding model: sentence-transformers/all-MiniLM-L6-v2.

**SelfCheckGPT:** Sample size $n \in \{5, 10, 15\}$, consistency metric: BERTScore with threshold $\tau \in \{0.6, 0.7, 0.8\}$.

**Calibrated Selective Classification:** Temperature scaling $T \in \{1.0, 1.5, 2.0\}$, Platt scaling regularization $\lambda \in \{0.01, 0.1, 1.0\}$.

**Entropy Methods:** Simple threshold tuning over validation set entropy values to achieve 2% target risk.

All methods calibrated on identical 500-sample splits with same train/val/test partitions to ensure fair comparison.

## M  ETHICS, SAFETY, AND REPRODUCIBILITY

### M.1  BIAS AND ABSTENTION GAMING

Certificates do not address underlying model biases. Abstention mechanism could be exploited via deliberate query crafting to induce abstention on sensitive topics. **Mitigation**: Deploy anomaly detection monitoring abstention patterns by topic, flagging significant shifts via KL-divergence thresholds.

### M.2  ALIGNMENT TENSION MITIGATION

RLHF reduces certifiability by collapsing model distributions. Joint training with certifiability rewards successfully preserves lift separability. Future work: integrate certifiability objectives into foundation model training. (See Figure 8 in main text for visualization.)

### M.3  SEVERITY VS FREQUENCY

A fundamental limitation of our approach, and selective classification more broadly, is that certificates control error frequency but not error severity. Our statistical guarantees ensure that at most $h^\star$ fraction of certified responses contain errors, but a factual mistake receives the same treatment as a catastrophic hallucination. This limitation is particularly concerning for medical, legal, or financial applications where error consequences vary dramatically. For high-stakes deployment, we recommend integrating with external harm detectors as secondary guardrails, developing severity-weighted risk metrics, or constraining deployment to lower-risk applications until severity-aware certification theory matures.

### M.4  CODE RELEASE AND EXPERIMENTAL SETUP

**Code Release:** Full implementation as 'certify-llm' library with sub-gamma parameter estimation, VSD optimization, and top-k compensation modules.

**Experimental Setup:** Random seeds (2025), exact prompts, skeleton templates, and hyperparameters documented. All results include 95% confidence intervals over 3 runs. Experiments on NVIDIA A100 GPUs ( 200 GPU hours total).

**Hyperparameters:** Clip $B = 12$, VSD $\lambda = 0.5$, batch size 64. Complete sweeps: $B \in \{8, 12, 16\}$, $\lambda \in \{0.1, 0.3, 0.5, 1.0\}$, batch sizes $\{16, 32, 64, 128\}$.

All hyperparameters including clip bound $B$, VSD tradeoff $\lambda$, and KL prior weight were selected using 5-fold cross-validation on the calibration set, maximizing coverage subject to meeting the 2% risk constraint. The reported results use hyperparameters selected on the validation set, with final evaluation on a held-out test set to prevent leakage. For baselines, we used identical validation procedures with the same calibration sets and risk targets to ensure fair comparison.

### M.5  BIAS AND FAIRNESS CONSIDERATIONS (EXTENDED)

While our certificates provide formal guarantees on selective risk, they may not address or could potentially amplify existing model biases. The choice of skeleton distribution is particularly critical, as biased skeletons could lead to systematically different abstention rates across different groups or topics.

### M.6  DEPLOYMENT CONSIDERATIONS

In real-world deployments, the abstention mechanism could have significant impacts on user experience and trust. High abstention rates might frustrate users, while low rates might not provide sufficient safety guarantees. Balancing these concerns requires careful consideration of the specific application context.

## M.7 LONG-TERM IMPLICATIONS

As AI systems become more prevalent in high-stakes decision making, certification mechanisms like ours may become regulatory requirements. This raises questions about standardization, auditing, and the potential for gaming or circumvention of safety measures.

Future research directions include extending the information lift framework to other modalities like vision or multimodal tasks, where skeletons might be based on simpler visual features or cross-modal baselines. Rather than using fixed skeletons, future work could explore dynamically constructing skeletons based on input context, potentially improving performance across diverse domains. Incorporating certifiability objectives directly into foundation model training could lead to models that are inherently more amenable to reliable uncertainty quantification. While our framework is robust to skeleton misspecification, a sophisticated adversary could attempt to attack the certificate directly by generating incorrect outputs intentionally crafted to have high information lift, suggesting the need for defenses such as ensemble skeletons or adversarial VSD training.

