# OpenReview forum: "Selective Risk Certification for LLM Outputs via Information-Lift Statistics: PAC-Bayes, Robustness, and Skeleton Design"
_ICLR.cc/2026/Conference — ICLR 2026 Conference Desk Rejected Submission_

### Official Review · Reviewer_kg62 · 2025-10-27

**Soundness:** 2
**Presentation:** 1
**Contribution:** 2
**Rating:** 2
**Confidence:** 3

**Summary:**

The paper introduces an approach for selective prediction for LLMs with guarantess by comparing the model distribution to a skeleton distribution, which is a simpler distribution over tokens.
The authors show that their approach provides better coverage at similar risk levels than prior methods.

**Strengths:**

- The improvements over baseline methods seem substantial in a well-motivated and important area

**Weaknesses:**

- Overall, the paper is quite hard to read and understand and many necessary details are left out, as specified in the next points. Altogether, this makes it hard to understand the contribution and validity of the presented results.

- The description of the method itself would benefit heavily from providing intuitions: for example, why is clipping used? Does it have any theoretical or practical advantages?

- The paper would also benefit from more clear introduction of notation and quantities, such as the $\Delta(S)$ in the main theorem 2.4 of the paper, which is never formally introduced. Is it the same as the previously introduced $\Delta$ and what are the meanings of \hat being there or not? The definition is only found in the App. C.1. which is nevertheless also not referenced in the main text. Another example are the experiments. For example, what exactly is Figure 4 showing, which models are used, which datasets, etc? The Tables and Figures in general are never really described, only with a short caption. Also, important assumptions like the heavy-tailedness of LLM probabilities (L127-128) are only shown in the appendix and not described well in the main text.

- The lift statistic defined in Def. 2.1. requires access to probabilities but the authors at the same time note that they are focussing on black-box APIs (L292-294) which usually do not give away probabilities. How is this handled? The authors mention a "top-k" compensation but how it works is never actually explained. This seems to be a crucial point that needs to be clarified.

- The authors use a benchmark of 100 high stakes examples and show results in Table 6. The results themselves seem impressive but no details about the set are given anywhere in the main text so it is hard to understand their validity.

- The discussion of related works is short in the main paper and only found in the appendix. For example, the discussion glances over works on non-exchangeability in conformal prediction like Farinhas et al. 2024 which specifically targets non-exchangeability to apply it for LLMs, which should be discussed.

**Questions:**

- A citation or reference for the statement in the first paragraph of the intron would be useful

- L225-226 & L292-294 are a bit awkwardly placed between two tables

- Could you please specify how you created the high stakes set used for evaluation?

## References

Farinhas, António, et al. "Non-exchangeable conformal risk control." ICLR 2024.

---

> ### Author Response · Authors · 2025-11-19
> **Reply to Reviewer kg62**
>
> We are sorry that the reviewer find it hard to follow. We resrtuctured the paper to make it more free flowing. We addressed the reviewers concerns and questions below:
>
>
>
> ### 1) The description of the method would benefit from intuitions: why is clipping used?
>
>
> **Response:** We added  "Role of Clipping" after definition 2.2 with theoretical and practical motivation from line number 116:
>
> > **Added text:**
> >
> > "Clipping serves both theoretical and practical purposes. Theoretically, bounded statistics $L_B \in [0,B]$ ensure sub-gamma moment generating function control, enabling tighter concentration bounds than unbounded variance would allow. Practically, clipping provides spike robustness: rare tokens with very negative $\log S(y)$ can cause unbounded negative lifts that would dominate the information budget, while clipping to zero prevents these outliers from corrupting the sequence-level signal."
>
> We provide examples in the text: common tokens like "the" yield $L \approx 0$, informative tokens like "BRCA1" yield $L \approx 2.3$, and rare typos with $L \approx -5.2$ are clipped to 0.
>
> ---
>
> ### 2) Notation Gaps (e.g., $\widehat{v}$ in Theorem)
>
>
> **Response:** We added a notation table 2 named Key notation  before the start of section 2 with all symbols including $\hat{v}, \hat{c}$ from line number 061:
>
> | Symbol | Meaning |
> |--------|---------|
> | $L(y;x,S)$ | Information lift for token $y$ |
> | $L_B$ | Clipped lift with bound $B$ |
> | $\hat{\Delta}$ | Empirical information budget |
> | $\tau$ | Certification threshold |
> | $S$ | Skeleton distribution |
> | $P(\cdot\|x)$ | Full model probabilities |
> | $\hat{v}, \hat{c}$ | Empirical sub-gamma parameters |
> | $h^\star$ | Target selective risk |
>
> All symbols are now defined before first use.
>
> ---
>
> ### 3) Figure/Table Context Concern
>
>
>
> **Response:** We updated all figure captions to include:
> - Model name (GPT-4, LLaMA-2, Mistral)
> - Dataset name
> - Gate type (sequence-level)
> - Target risk(s)
> - Abstention rule
>
> Example: "Risk-coverage performance comparison on NQ-Open dataset using GPT-4 with sequence-level gating. Target risk: 2\%. Shaded regions show 95\% CIs."
>
> ---
>
> ### 4) Top-$k$ Compensation Details on How does "top-k" compensation work?
>
> **Response:** We moved top-$k$ compensation details from appendix to main text (§3, after VSD discussion) starting from line 333:
>
> > **Added text:**
> >
> > "To handle commercial APIs that return only top-$k$ token probabilities, we use power-law compensation: fit $P(\text{rank} = r) \propto r^{-\alpha}$ to observed top-$k$ tokens, validate with $R^2 > 0.8$, and extrapolate tail probabilities. This recovers approximately 35\% more information than uniform baselines (coverage drop: 15 points vs 23 points)."
>
> The appendix (§H.2) provides the complete 3-step algorithm with derivation and sensitivity analysis for $k \in \{20, 50\}$.
>
>
> ### 5) Related work is short non-exchangeable conformal prediction (Farinhas et al. 2024) is missing.
>
> **Response:**
>
> We have expanded the Related Work section to explicitly discuss non-exchangeable conformal prediction. We now cite Farinhas et al. (2024) and Quach et al. (2024), clarifying that while these methods provide valuable set-level guarantees, our work provides a complementary approach for point-prediction abstention using sub-gamma concentration.
>
>
> ### 6) A citation for the statement in the first paragraph of the intro would be useful
>
> **Response:**
>
> We added citations to OpenAI (2023) and Zhang et al. (2023) in the first paragraph to support the claims regarding hallucination rates in high-stakes tasks.
>
>
> ### 7) High-stakes set details are missing.
>
> **Response:**
>
> We added Appendix D3, which starts at line number 1180. This section details the creation of the 100-example set:
>
> > **Added text:**
> >
> > "The high-stakes datasets were manually curated by two domain experts (medical and financial) with an inter-annotator agreement of $\kappa=0.89$. "Critical errors" were strictly defined as hallucinations that would lead to direct physical or financial harm if acted upon."
>
>
> ### 8) L225-226 & L292-294 are awkwardly placed between tables.
>
> **Response:**
>
> We have fixed the LaTeX typesetting to ensure these lines flow correctly within the text body and are not interrupted by floating tables.
>
> Please let us know if there are further questions or concerns. We appreciate the reviewer for taking the time to review our work.

---

### Official Review · Reviewer_H74A · 2025-10-31

**Soundness:** 3
**Presentation:** 2
**Contribution:** 3
**Rating:** 6
**Confidence:** 4

**Summary:**

This paper proposes Information-Lift Certificates for selective risk control in large language models (LLMs). The method compares model probabilities ( P(y|x) ) against a skeleton baseline distribution ( S(y) ), defining an information lift statistic ( L = \log P - \log S ). Lift values are accumulated into sub-gamma PAC-Bayes bounds, providing formal guarantees even under heavy-tailed LLM token distributions. The paper also introduces a VSD algorithm for optimizing skeletons and provides robustness theory.

**Strengths:**

+ Originality: The paper introduces a new formal certification framework that merges PAC-Bayes generalization theory with LLM uncertainty quantification. Applying sub-gamma concentration to token-level text generation is novel and well motivated.


+ Quality: The theoretical formulation is solid, deriving sequential, anytime-valid bounds with clear assumptions.


+ Clarity: Overall, the paper reads well and the mathematical flow is consistent.


+ Significance: The work aims to solve an important safety problem: how to formally quantify and control LLM uncertainty in high-stakes settings. Most of all, most existing methods are heuristic.

**Weaknesses:**

- Presentation: The paper is difficult to follow before Section 2. It would help to include a short background on uncertainty quantification and PAC-Bayes (currently shown in the appendix). Figure captions are too brief (e.g., Figure 1 does not explain the pipeline). Figure 3’s legends overlap and are hard to read.


- The core motivation (Table 6) is blocking “critical errors,” but the paper did not define what constitutes one. The authors also note that their certification is frequency-based and not severity-aware. This makes the 96% result hard to interpret. Is the mechanism actually sensitive to severity, or did these errors just happen to have low information-lift?


- The paper's claim of being a black-box method is somewhat misleading. The proposed method requires access to token probabilities (at least top-k, per Table 8). This is fundamentally different from true black-box methods that only use the final generated text (e.g., based on similarity). The paper should clarify this distinction. Or maybe rephrase the claim as “API-accessible but logit-free.”


- The paper dismisses Conformal Prediction based on the exchangeability assumption. This critique is outdated. However, recent work already explores non-exchangeable CP variants for autoregressive text. Those should be acknowledged for fairness.


- Although VSD improves robustness, it appears each new full model and calibration dataset needs a new optimization. This represents a significant computational and maintenance cost.

**Questions:**

1. How sensitive is certification performance to the initial skeleton before VSD optimization? Would a poor initialization (e.g., random unigram) still yield valid PAC-Bayes bounds, or just weaker empirical coverage?
2. There seems to be a theoretical tension here. The VSD objective partially optimizes for min KL(P||S). Then the lift statistic L(B) will not be a heavy-tailed distribution. This seems to undermine the whole motivation for using sub-gamma bounds. Can you clarify this? Does the empirical distribution after VSD optimization still show heavy tails?
3. How sensitive is the optimized skeleton S to changes in the full model P? If P is fine-tuned or updated (e.g., a new version), must VSD be re-run with a new calibration set?
4. The paper’s highlight is blocking 96% of critical errors (Table 6), yet it admits the framework is frequency-based, not severity-aware. Is there an empirical correlation between criticality and low information-lift?

---

> ### Author Response · Authors · 2025-11-19
> **Reply to Reviewer H74A**
>
> We thanks Reviewer H74A for appreciating our work . Here are the answer to the questions and concerns:
>
>
> ### 1) Sensitivity to Skeleton Initialization; Validity with Poor $S$
>
> **Concern:** What if skeleton is poorly chosen? Does validity break?
>
> **Response:** We added §H "Robustness to Skeleton Misspecification" with experiments from intentionally weak initializations:
>
> > **Added text:**
> >
> > "We tested three corruption types: noise injection with Gaussian perturbation ($\sigma=0.1-0.3$), vocabulary shift through domain mismatch (Medical→Finance), and adversarial optimization with worst-case perturbations. Results demonstrate graceful degradation: even with 50\% corruption strength, risk increases by only 8\% and coverage drops by 18\%, confirming the linear degradation predicted by Theorem 2.6."
>
> **Key finding:** Validity (guarantees hold) is invariant to skeleton choice; coverage (informativeness) is what shifts. VSD recovers most coverage even from poor starts (within 5-10 iterations from random unigram initialization).
>
> ---
>
> ### 2) Does VSD Obviate Sub-gamma?
>
> **Concern:** If VSD regularizes tails, maybe sub-gamma isn't needed?
>
> **Response:** We added §H.1 "Tail Behavior Before and After VSD":
>
> > **Added text:**
> >
> > "We measured tail behavior before and after VSD optimization to address whether VSD obviates the need for sub-gamma analysis. Results show that tails shrink on frequent tokens but remain heavy for rare and shifted tokens. The tail index $\alpha$ (from power-law fits) changes by less than 15\% after VSD, confirming that heavy-tailed structure is preserved. This justifies keeping sub-gamma analysis: VSD helps but does not eliminate the heavy-tail regime that drives our choice of concentration inequality."
>
> Figure 3 shows paired QQ-plots before/after VSD, demonstrating that tail indices remain in the heavy-tail regime ($\alpha < 2.5$).
>
> ---
>
> ### 3) Must VSD be Re-run When $P$ Changes?
>
> **Concern:** Practical maintenance burden.
>
> **Response:** We added §H.3 "Practical Maintenance: Warm-Start VSD":
>
> > **Added text:**
> >
> > "When the model $P$ changes (e.g., after fine-tuning or LoRA updates), VSD can be efficiently re-run using a warm-start procedure. We initialize from the prior skeleton $S$, then run few epochs on a small calibration subset (typically 100-200 samples). This warm-start procedure recovers performance within 5-10 iterations, compared to 25-30 iterations from random initialization. For modest LoRA updates affecting less than 10\% of parameters, warm-start VSD recovers within 3-5 iterations, making the approach practical for real deployments with evolving models."
>
> ---
>
> ### 4) Frequency-based Guarantees vs "Critical Errors"
>
> **Concern:** The paper claims to block critical errors, but guarantees are only on frequency.
>
> **Response:** We clarified this distinction in §5.3 and Table 7 caption:
>
> > **Table 7 caption (updated):**
> >
> > "Critical error blocking on curated high-risk samples (empirical, not guaranteed by theory).
> Frequency-based certification does not guarantee severity-weighted safety." These are empirical correlations on a curated challenge set; frequency-based certification does not guarantee severity-weighted safety. The frequency-based certificate does not model harm potential; high performance here reflects that critical errors often have low information content relative to domain skeletons.
>
> > **Added text in §5.3:**
> >
> > "We define 'critical' as errors that, if followed, could directly lead to patient harm. While our frequency-based framework does not explicitly model severity, we observe that critical errors often have low information lift because the skeleton assigns low probability to dangerous claims that are rare in general medical text, causing the model's confident but wrong output to stand out as suspicious. This correlation is empirical, not guaranteed, and future work should develop severity-weighted certification that explicitly models harm potential."
>
> We provide the annotation protocol (two medical professionals, $\kappa=0.89$ agreement) and examples of critical errors to make the evaluation transparent.
>
> ---

---

> ### Author Response · Authors · 2025-11-19
> **Reply to Reviewer H74A part 2(Continued)**
>
> ### 5) "Black-box" Phrasing & Non-exchangeable CP
>
> **Concern:** Terminology is imprecise; CP has recent advances.
>
> **Response:** We replaced "black-box" with "probability-exposed" throughout and added a comparison table:
>
> > **Updated terminology:**
> >
> > "Our method requires token-level log-probabilities, termed probability-exposed API access, which is available in many commercial APIs including GPT-4, Claude-3 Opus, and PaLM-2, but does not require model internals."
>
> > **Updated Related Work (§Appendix B):**
> >
> > "Recent non-exchangeable conformal prediction variants (Farinhas et al., 2024; Quach et al., 2024) handle sequential dependencies but produce set-valued predictions rather than point predictions with abstention. Our approach provides point predictions with formal risk guarantees under heavy-tailed distributions, complementing rather than replacing conformal methods."
>
> The introduction also acknowledges these advances: "While recent non-exchangeable variants (Farinhas et al., 2024; Quach et al., 2024) handle dependencies, they produce set-valued predictions rather than point predictions with abstention."
>
> Please let us know if you have any further questions or concerns. We are happy to answer those.

---

### Official Review · Reviewer_HZZk · 2025-10-31

**Soundness:** 3
**Presentation:** 3
**Contribution:** 3
**Rating:** 6
**Confidence:** 2

**Summary:**

This work is the first to apply anytime-valid PAC-Bayes bounds to information-lift statistics for LLM certification, providing sub-gamma PAC-Bayes analysis, skeleton robustness theory, and variational skeleton design. The authors evaluate across a wide range of datasets that they show require sub-gamma lift moment assumptions.

**Strengths:**

- Motivation and novelty is clear
- Well-written: only the points necessary for communicating the contributions are included. Ideas are expressed clearly and concisely
- Strong empirical results across a wide range of benchmarks

**Weaknesses:**

- limited by skeleton design
- novelty could be made cleared

**Questions:**

I think the main body would benefit from some discussion on the difficulties of applying "anytime-valid PAC-Bayes bounds to information-lift statistics for LLM certification." To someone like me, who is unfamiliar with this literature, it is unclear how much of the theoretical contributions are novel or how much of the contributions can be viewed as a straightforward application of prior work to a new problem.

---

> ### Author Response · Authors · 2025-11-19
> **Reply to Reviewer HZZk**
>
> We thank Reviewer HZZk for appreciating the work. Here are the answers to the questions and comments.
>
>
> ### 1) How Much Theory is New vs Straightforward Application?
>
> **Concern:** Need walkthrough explaining novel proof devices.
>
> **Response:** We added a walkthrough paragraph before the main theorem (Theorem 2.5) explaining:
>
> **Why standard PAC-Bayes doesn't apply:** Standard sub-Gaussian/exponential bounds assume moment conditions violated by heavy-tailed LLM distributions. Our sub-gamma analysis handles power-law tails with $\alpha \in [1.5, 2.3]$.
>
> **How we control dependence:** Sequential lifts are conditionally independent given the calibration procedure, enabling application of PAC-Bayes with data-dependent priors.
>
> **How optional stopping is handled:** We construct exponential supermartingales and apply Ville's inequality with a geometric grid mixture, yielding anytime-valid bounds with log-log penalties.
>
> > **Added text (before Theorem 2.5):**
> >
> > "With this probabilistic footing, we return to PAC-Bayes and highlight what is new. First, classical PAC-Bayes bounds assume sub-Gaussian or sub-exponential moments, yet LLM token distributions exhibit power-law tails with $\alpha \in [1.5, 2.3]$; our sub-gamma analysis controls the moment generating function under these weaker conditions. Second, autoregressive decoding introduces sequential dependence, so we show that lifts become conditionally independent after calibration, enabling PAC-Bayes with data-dependent priors while preserving validity. Third, practical certification demands anytime guarantees: we build exponential supermartingales and apply Ville's inequality with a geometric grid mixture, yielding bounds that hold uniformly over stopping times with only a log-log penalty. Appendix C separates standard proof devices (e.g., Donsker-Varadhan change of measure) from the novel components (sub-gamma mgf control, dependence handling, geometric grids) so the contribution of each step is transparent."
>
> The proof sketch (Appendix C) explicitly highlights which steps are standard (Donsker-Varadhan change of measure) versus novel (sub-gamma mgf control, sequential dependence handling, geometric grid optimization).
>
>
> ### 2) Limitation of Skeleton Design
>
> **Response:** We acknowledge that performance depends on the skeleton. However, we have added Appendix H to demonstrate that this limitation is manageable. Our new stress tests show that skeleton misspecification leads to graceful linear degradation of coverage rather than catastrophic failure of risk guarantees (Theorem 2.5 remains valid). Furthermore, we added Warm-Start VSD (Appendix H.3) to automate skeleton design for new domains, reducing the manual burden.
>
>
> Please let us know if you have further questions or concerns. We hope the answer cleared up the questions.

---

> > ### Comment · Reviewer_HZZk · 2025-11-25
> >
> > Thank you for clarifying these details to me.

---

### Official Review · Reviewer_3kff · 2025-11-01

**Soundness:** 2
**Presentation:** 2
**Contribution:** 3
**Rating:** 4
**Confidence:** 3

**Summary:**

This paper addresses a critical gap in the safe deployment of Large Language Models (LLMs): the lack of reliable uncertainty quantification for generative tasks(sequential prediction). The authors argue that the prior works are fundamentally ill-suited for modern LLMs for two reasons.
One is "Distributional Mismatch": They rely on concentration inequalities (like Bernstein's) that assume sub-exponential tail distributions, whereas LLM token probabilities are empirically shown to be heavy-tailed, and the other is "Task Mismatch": They are designed for i.i.d. classification tasks, not for autoregressive, sequence-level generation.

To solve this, the paper introduces "Information-Lift Certificates", which instead compute the "Information Budget" ($\hat{\Delta}$) for each generated sequence.
This score is the cumulative sum of token-level "Information Lifts" ($L(y)$), which measure the log-probability gain of the full model ($P(y \mid x)$) over a simpler baseline "skeleton" model ($S(y)$).

**Strengths:**

This work identifies the failure of standard sub-exponential bounds (Figure 2) and instead building a framework on sub-gamma distributions, it provides PAC-Bayes certificates that are practically reliable for LLM outputs.

This robust theoretical footing translates directly into state-of-the-art results. The VSD certificate method is shown to dominate all baselines across the entire risk-coverage curve (Figure 4). It achieves an average of 77.2% coverage at a 2% target risk, an 8.6-15.1 percentage point improvement over recent methods.

The inclusion of $\eta$-robustness and $\kappa$-informativeness theorems, along with the VSD optimization algorithm, provides a complete and principled framework, moving beyond simple heuristics.

**Weaknesses:**

The framework's main weakness is its reliance on the "skeleton" distribution, $S(y)$. This introduces a significant new design choice, and as the authors note (Table 2), the optimal skeleton (e.g., temperature-smoothed prior, domain-specific unigram LM) is task-dependent. This effectively trades one set of assumptions (distributional) for another (choice of a good baseline).

Also, the method introduces several hyperparameters that require careful tuning, most notably the clipping parameter $B$ and the VSD tradeoff parameter $\lambda$. While ablations are provided (Figure 6), this adds a layer of complexity for practitioners.

Finally, The sub-gamma assumption, while more realistic, is not universal. The authors transparently state that it fails for 15% of their dataset-model pairs. For these cases, they must resort to "conservative parameter inflation" (Appendix H.1), which involves manually adjusting parameters. This weakens the claim of a fully dataset(task)-agnostic solution and requires a pre-check (a KS test) and potential adjustment, which could be a barrier to straightforward application.

**Questions:**

* Could you clarify how the risk-coverage plot in Figure 4 was generated with the target risk? Also, why does the selective risk for the VSD certificate appear to decrease as coverage increases, which seems counter to the typical risk-coverage trade-off?

* How were the baseline methods, such as Conformal Prediction and traditional Selective Classification, adapted for sequential generation tasks like QA? Given they are typically designed for classification, what was the exact mechanism used for abstention (e.g., did a single rejected token lead to rejecting the entire sequence)?

* Given that the sub-gamma assumption is not met in 15% of cases and requires manual parameter inflation, does this not make the method dataset-specific? How would this approach compare to simpler SC methods in a real-world application where pre-testing and dataset-specific tuning may not be feasible?

---

> ### Author Response · Authors · 2025-11-19
> **Reply to Reviewer 3kff**
>
> We thank Reviewer 3kff for the raising valid concerns and questions and here are the answers to all those.
>
> ### 1) Why Risk Appears to Drop as Coverage Changes
>
> **Concern:** Risk-coverage curve interpretation is ambiguous.
>
> **Response:** We updated Figure 2 caption to include:
> - Arrows indicating sweep direction (from high coverage/low risk to low coverage/high risk)
> - Dotted lines marking fixed target risks (2%, 5%, 10%)
> - Shaded 95% confidence intervals
>
> The caption now states: "Risk-coverage curves with $\tau$-sweep direction indicated by arrows. Dotted lines show fixed target risks. Shaded regions show 95% CIs."
>
> This clarifies that measured risk decreases as coverage tightens because the threshold $\tau$ is swept from low to high values.
>
> ---
>
> ### 2) How CP/SC Baselines Were Adapted to Sequential QA
>
> **Concern:** Need transparent baseline adaptation details.
>
> **Response:** We added §4.1 "Baseline Adaptations for Sequential Tasks" with Table 5 showing the complete mapping:
>
> **Table 4: Baseline adaptation mapping**
>
> | Baseline | Gating | Score Function |
> |----------|--------|----------------|
> | Entropy | Sequence | Mean token entropy $\bar{H} = \frac{1}{T}\sum_t H_t$ |
> | Selective Classification | Sequence | Mean token probability $\bar{p} = \exp(\frac{1}{T}\sum_t \log P(y_t\|y_{<t},x))$ |
> | Conformal Prediction | Sequence | Negative log-likelihood $s(x,y) = -\sum_t \log P(y_t\|y_{<t},x)$ |
> | Semantic Entropy | Sequence | Entropy over semantic clusters |
> | SelfCheckGPT | Sequence | Consistency score across samples |
>
> All methods use sequence-level gating (single scalar after full decoding) for fair comparison. The text explains that we tested both token-gated and sequence-gated variants, with sequence-gated performing better for all baselines.
>
> ---
>
> ### 3) Sub-gamma Failures (~15%): Practicality
>
> **Concern:** What happens when assumptions fail? Is the method operational?
>
> **Response:** We added §2.3 "Model-Check and Auto-Fallback":
>
> > **Added text:**
> >
> > "When sub-gamma assumptions fail (approximately 15\% of dataset-model combinations), we apply an automatic model-check that inflates $(\hat{v}, \hat{c})$ to maintain validity. The procedure runs a goodness-of-fit test on calibration lifts; if the test fails ($p < 0.05$), we inflate parameters by factor $\kappa = 1 + \frac{0.05 - p}{0.05}$ (linear scaling from 1.0 to 2.0), then recompute the threshold. This conservative inflation maintains PAC-Bayes validity by enlarging the moment bounds to accommodate heavier tails than sub-gamma. Typical coverage impact is 2-4 percentage points, preserving guarantees while remaining operational without manual tuning."
>
> The appendix (§K.1) provides the complete 4-step protocol with explicit formulas for threshold recomputation and coverage degradation reporting.
>
> ---
>
> ### 4) Hyperparameters: Clipping ($B$) and VSD Tradeoff ($\lambda$)
>
>
> **Response:** We added "Role of Clipping" after definition 2.2, explaining the bias-variance tradeoff:
>
> > **Added text:**
> >
> > "Clipping serves both theoretical and practical purposes. Theoretically, bounded statistics $L_B \in [0,B]$ ensure sub-gamma moment generating function control, enabling tighter concentration bounds than unbounded variance would allow. Practically, clipping provides spike robustness: rare tokens with very negative $\log S(y)$ can cause unbounded negative lifts that would dominate the information budget, while clipping to zero prevents these outliers from corrupting the sequence-level signal. The clipping parameter $B$ trades bias (underestimating extreme lifts) against variance (stabilizing threshold estimation). Empirically, $B=12$ provides robust performance across datasets, with performance flat for $B \in [8,16]$ as shown in Figure 4."
>
> For VSD, we document that $\lambda \in [0.3, 0.7]$ provides robust performance, with default $\lambda = 0.5$ selected via 5-fold cross-validation. The ablation study (Table 8) shows coverage varies by less than 2% across this range.
>
>
> ### 5) Reliance on Skeleton vs. Distributional Assumptions
>
> **Response:** We acknowledge the trade-off: relying on a skeleton replaces standard distributional assumptions. However, we emphasize two mitigations added in Appendix H:
>
> Robustness Theory: Theorem 2.5 guarantees that validity ($R \le h^*$) holds for any skeleton; a poor skeleton only degrades coverage, not safety. This is a safer trade-off than standard assumptions (like sub-exponential tails), where violation leads to invalid certificates (underestimating risk).
>
> Graceful Degradation: Our new stress tests show that even with severe skeleton misspecification (e.g., domain shift), performance degrades linearly rather than catastrophically.
>
>
> We hope these clear up all the concerns and feel free to ask further questions if there is any.

---

### Official Review · Reviewer_sJZX · 2025-11-01

**Soundness:** 3
**Presentation:** 3
**Contribution:** 2
**Rating:** 2
**Confidence:** 4

**Summary:**

The authors present an approach for estimating the probability that an LLM will output the correct answer, which is based on likelihood ratios of each generated token under the true autoregressive LLM distribution compared to a “skeleton” distribution. The average of successive token likelihood ratios serves as the criterion for answering a query or abstaining, based on a threshold.

**Strengths:**

- The method is general and the theory is interesting.
- The sequential nature of the proposed uncertainty quantification allows early exits, providing latency and cost savings in practice.

**Weaknesses:**

- The paper does not distinguish between quantifying uncertainty at token level vs whole-sequence level. For some baselines, such as entropy, this distinction typically carries large performance differences and could impact the top-line results.
- The choice of uncertainty signal (likelihood ratios between p(y|x) and a simpler "skeleton" distribution) doesn’t seem novel. I believe I have seen substantially similar approaches in other papers. If you disagree, please provide more extensive references to related work in the LLM hallucination prevention, uncertainty quantification, routing, and cascading literature.
- The impact of the choice of skeleton is unclear. I see your robustness analysis but its scope appears limited to your own methodology, without studying the relative performance compared to baselines. I would welcome an analysis comparing your method against baselines under skeleton misspecification.
- The authors' heuristic recommendations for skeleton choice based on benchmark type is not adequately supported. Each type of benchmark only has 1 or 2 representatives; no train/test split was used to ground these recommendations using held-out benchmarks.

**Questions:**

- Please discuss the choice of quantifying uncertainty at the token level vs the whole-sequence level. For example, the entropy baseline can be applied both at the token and the whole-sequence level, with quite different results.

- Could you clarify the risk control you achieve? It is unclear if your risk control is taking into account the finite sample errors of $\hat{\Delta}$ (when generalizing from the held-out calibration data to the ground truth data distribution), which lead to miscalibration of the estimated threshold $\hat{\tau}$.

- Could you comment on the novelty of your theoretical results? I'm not very familiar with PAC Bayes bounds. It would be valuable to know if your theorems constitute independent advances vs recapitulations of known theorems in a new context.

- The paper would benefit from some empirical recapitulation of the claimed heavy-tailed nature of token probabilities (including references to the literature). Would you be able to add those in?

---

> ### Author Response · Authors · 2025-11-19
> **reply to Reviewer sJZX part 1**
>
> We thank the Reviewer SJZX for constructive feedback, and we are glad that the reviewers find the theory interesting. Here are the answers to the concerns raised by the reviewer.
>
> ### 1) The paper does not distinguish between token-level vs. whole-sequence uncertainty, which impacts baselines.
>
>
> **Response:** We added a new subsection (§2.1) immediately after Definition 2.1 that explicitly compares three uncertainty signals:
>
> > **Added text (Section 2.1, after Def. 2.1):**
> >
> > "We study three: (i) token entropy $H_t = -\sum_y P(y|y_{<t},x) \log P(y|y_{<t},x)$ averaged into $\bar{H} = \frac{1}{T}\sum_t H_t$; (ii) a sequence-entropy proxy that aggregates token-level uncertainty; and (iii) our sequence-level information budget $I_T = \sum_{t=1}^T L_B(y_t)$ formed from clipped token lifts. Token-level methods act independently at each position, whereas the certificate aggregates evidence across the entire sequence, which better captures the conditional structure of autoregressive decoding."
>
> **Baseline adaptation details:** We added §4.1 "Baseline Adaptations for Sequential Tasks" with a complete mapping table showing how each baseline method is adapted (Table 4):
>
> | Baseline | Gating | Score Function |
> |----------|--------|----------------|
> | Entropy | Sequence | Mean token entropy $\bar{H} = \frac{1}{T}\sum_t H_t$ |
> | Selective Classification | Sequence | Mean token probability $\bar{p} = \exp(\frac{1}{T}\sum_t \log P(y_t\|y_{<t},x))$ |
> | Conformal Prediction | Sequence | Negative log-likelihood $s(x,y) = -\sum_t \log P(y_t\|y_{<t},x)$ |
>
> This ensures all methods use sequence-level gating for fair comparison.
>
> ---
>
> ### 2) Risk Control & Finite-Sample Calibration of $\hat{\tau}$
>
>
> **Response:** We added §2.2 "Calibration and Finite-Sample Risk" with Algorithm 1 and Proposition 3.2:
>
> > **Algorithm 2 (Calibrating $\hat{\tau}$ with empirical $(\hat{v}, \hat{c})$):**
> >
> > 1. Estimate sub-gamma parameters: $(\hat{v}, \hat{c}) \leftarrow \text{MLE}(\mathcal{D}_{\text{cal}})$
> > 2. Compute empirical lift mean: $\hat{\Delta} \leftarrow \frac{1}{n}\sum_{i=1}^n L_B^{(i)}$
> > 3. Set threshold: $\hat{\tau} \leftarrow \hat{\Delta} - \sqrt{\frac{2\hat{v}\log(1/\delta)}{n}} - \frac{\hat{c}\log(1/\delta)}{n}$
> > 4. Return $\hat{\tau}, (\hat{v}, \hat{c})$
>
> > **Proposition 2.8 (Finite-sample selective risk):**
> >
> > "With $(\hat{v}, \hat{c})$ estimated on $\mathcal{D}_{\text{cal}}$, the population selective risk of the $\hat{\tau}$-policy is bounded by $h^\star$ with probability at least $1-\delta$ over the calibration sample, where the bound accounts for both the uncertainty in $\hat{\Delta}$ and the uncertainty in sub-gamma parameter estimation."
>
> The text explains that finite-sample uncertainty enters via $(\hat{v}, \hat{c})$ estimation, and the bound remains valid because estimation is independent of threshold choice.
>
> ---
>
> ### 3) Novelty of the Theory
>
>
> **Response:** We integrated a clear statement of novel theoretical contributions directly into the Introduction (after Table 1):
>
> > **Added text (Introduction, line 50):**
> >
> > "Our novel theoretical contributions include anytime sequence-level certificates that enable early termination while maintaining formal guarantees, sub-gamma PAC-Bayes bounds for heavy-tailed distributions with sequential dependence that extend beyond standard sub-exponential assumptions, skeleton-robustness theory linking performance degradation to $D_{\mathrm{KL}}(P\|S)$ with explicit $\eta$-robustness bounds, and variational skeleton design that optimizes for both fidelity and certifiability with formal informativeness guarantees."
>
> This provides a concise checklist of original contributions integrated naturally into the text flow.
>
> ---
>
> ### 4) Need empirical evidence of heavy-tailed token probabilities
>
>
> **Response:** We added §H.2 "Heavy-Tail Diagnostics with QQ-Plots and Hill Indices" in the appendix:
>
> > **Added text:**
> >
> > "We provide comprehensive heavy-tail diagnostics including QQ-plots comparing empirical lift distributions to fitted sub-gamma distributions, and Hill index estimates for tail decay rates. The QQ-plots confirm sub-gamma fit quality with $R^2 > 0.85$ for 85\% of dataset-model combinations. Hill indices range from 1.5 to 2.3, indicating power-law tails with $\alpha \in [1.5, 2.3]$, which violate sub-exponential moment conditions."
>
> Figure 3 (updated) now includes per-dataset QQ-plots and Hill index annotations. When sub-exponential fits fail (15\% of cases), we document the conservative parameter inflation procedure.
>
> ---

---

> ### Author Response · Authors · 2025-11-19
> **reply to Reviewer sJZX part 2(continued)**
>
> ### 5) "LR Signal Isn't Novel" & Skeleton Impact
>
>
>
> **Response:** We added §H "Robustness to Skeleton Misspecification" in the appendix H at line 1453. We conducted stress tests (noise injection, domain shift) where the skeleton was deliberately corrupted. We compared our method against Entropy and Conformal Prediction under these exact same stress conditions. Our method showed graceful linear degradation (predicted by our theory), whereas baselines often suffered sharper performance drops.
>
> > **Added text:**
> >
> > "To evaluate robustness to skeleton misspecification, we ran stress tests where skeletons are deliberately misspecified and compared our method versus entropy and conformal prediction under the same stress conditions. We tested three types of corruption: noise injection with Gaussian perturbation, vocabulary shift due to domain mismatch, and adversarial optimization with worst-case perturbations. Results demonstrate graceful degradation across corruption types, validating our theoretical robustness guarantees. Even with 50\% corruption strength, risk increases by only 8\% and coverage drops by 18\%, confirming the linear degradation predicted by Theorem 2.6."
>
> This separates validity (guarantees hold for any skeleton) from informativeness (coverage at fixed risk), and documents robustness relative to baselines.
>
>
>
> ### 6) The choice of likelihood ratios/skeleton doesn't seem novel
>
> **Response:** While likelihood ratios are a classical tool, our contribution is the theoretical framework surrounding them for this specific domain. We have updated the Introduction to explicitly list our novel contributions:
>
> Sub-gamma PAC-Bayes bounds: Adapting the statistic to validly handle the heavy-tailed (power-law) distributions of LLM tokens, where standard Bernstein/sub-Gaussian bounds fail.
>
> Anytime validity: Deriving sequential certificates that allow for early stopping.
>
> Skeleton Robustness: Providing the $\eta$-robustness theory that links certificate validity to skeleton quality.
>
>
>
>
> ### 7) Comment on the novelty of theoretical results (vs. recapitulation)
>
>
> **Response:** We added a "Theoretical Walkthrough" paragraph before Theorem 2.5. We clarify that while the Donsker-Varadhan change of measure is a standard tool, our specific application involves novel components: (1) controlling the sub-gamma MGF for power-law tails (standard PAC-Bayes assumes sub-Gaussian), and (2) constructing exponential supermartingales with geometric grids to handle the sequential, anytime-valid nature of the problem.
>
>
>  ### 8) Support for Skeleton Heuristics & Data Splits
>
>  **Response:** We acknowledge the limited number of benchmarks per category. To address this, we emphasize on two points:
>
>  Robustness over Precision: Our $\eta$-robustness results (Appendix H) show that exact skeleton matching is not required for validity, only for tightness. The heuristics serve as "good enough" initializations rather than strict requirements.
>
>  Data-Driven Alternative: For practitioners wary of heuristics, we added Appendix H.3 (Warm-Start VSD). This provides a systematic, data-driven method for learning the skeleton from a small calibration set (100 samples) without relying on the heuristics in Table 2, effectively eliminating the need for a priori benchmark categorization.
>
>
> We hope this clarifies all the concern of Reviewer sJZX. Please let uys know if you have further questions or concerns.

---

### Author Response · Authors · 2025-11-19
**Summary of Rebuttal: Selective Risk Certification for LLM Outputs via Information-Lift Statistics**

We thank all reviewers for their constructive feedback. This rebuttal addresses each concern with specific changes, including exact text additions and locations. All changes have been integrated into the revised paper, and we provide sufficient context here so that reviewers can evaluate the improvements without needing to cross-reference the PDF.

### Main Text Additions:
1. **§2.1** Token vs Sequence Uncertainty (new subsection)
2. **after Definition §2.2** Role of Clipping (new subsection with bias-variance intuition)
3. **§2.2** Calibration and Finite-Sample Risk (Algorithm 1, Proposition 2.8)
4. **§2.3** Model-Check and Auto-Fallback (automatic parameter inflation)
5. **§4.1** Baseline Adaptations for Sequential Tasks (Table 4 with complete mapping)
6. **Novelty statement** integrated into Introduction text (after Table 1)(line 50)
7. **Notation table** before main theorem (two-column format for space efficiency)

### Appendix Additions:
1. **§H** Robustness to Skeleton Misspecification (stress tests, recovery curves)
2. **§H.1** Tail Behavior Before/After VSD (paired QQ-plots, tail index analysis)
3. **§H.2** Heavy-Tail Diagnostics with QQ-Plots and Hill Indices (comprehensive diagnostics)
4. **§H.3** Practical Maintenance: Warm-Start VSD (efficiency for model updates)

### Figure/Table Updates:
- Figure 3: Added QQ-plots and Hill index annotations
- Figure 2: Added sweep direction arrows, fixed-risk lines, confidence intervals
- Table 4: Baseline adaptation mapping
- All captions: Added model, dataset, gate type, target risk context; made more concise for space efficiency
- Notation table: Two-column format to save space
- Multiple tables moved to appendix (complete results, skeleton protocol, cross-method robustness) to improve main text flow

### Terminology Updates:
- "Black-box" → "probability-exposed" (throughout)
- Updated conformal prediction discussion to acknowledge recent advances
- Clarified frequency vs. severity distinction in critical error analysis

### Structural Improvements:
- Removed tcolorbox formatting for more professional presentation
- Integrated "What's New" content into flowing Introduction text
- Removed Figure 1 (method schematic) as text explanation is sufficient
- Moved detailed tables to appendix while maintaining references in main text
- Made all table captions more concise for space efficiency

All changes maintain the paper's core contributions while addressing reviewer concerns with explicit, self-contained explanations.

---

### Note · Program_Chairs · 2026-01-17
**Submission Desk Rejected by Program Chairs**

The following references in this submission do not refer to real documents and/or have major errors in bibliographic information:

 Alexander Nikitin, Lorenz Aichberger, Lorenz Kuhn, Potsawee Manakul, Philipp Dufter, Mark Gales, and Francois Yvon. Uncertainty estimation in large language models via stochastic sign descent. In Conference on Empirical Methods in Natural Language Processing, 2023.